# The Effect of Carbon Nanofibers on the Mechanical Performance of Epoxy-Based Composites: A Review

**DOI:** 10.3390/polym16152152

**Published:** 2024-07-29

**Authors:** Paulo Santos, Abílio P. Silva, Paulo N. B. Reis

**Affiliations:** 1C-MAST—Centre for Mechanical and Aerospace Science and Technologies, University of Beira Interior, 6201-001 Covilhã, Portugal; abilio@ubi.pt; 2University of Coimbra, CEMMPRE, ARISE, Department of Mechanical Engineering, 3030-788 Coimbra, Portugal

**Keywords:** polymer composites, epoxy matrix, carbon nanofibers, mechanical properties

## Abstract

This review is a fundamental tool for researchers and engineers involved in the design and optimization of fiber-reinforced composite materials. The aim is to provide a comprehensive analysis of the mechanical performance of composites with epoxy matrices reinforced with carbon nanofibers (CNFs). The review includes studies investigating the static mechanical response through three-point bending (3PB) tests, tensile tests, and viscoelastic behavior tests. In addition, the properties of the composites’ resistance to interlaminar shear strength (ILSS), mode I and mode II interlaminar fracture toughness (ILFT), and low-velocity impact (LVI) are analyzed. The incorporation of small amounts of CNFs, mostly between 0.25 and 1% by weight was shown to have a notable impact on the static and viscoelastic properties of the composites, leading to greater resistance to time-dependent deformation and better resistance to creep. ILSS and ILFT modes I and II of fiber-reinforced composites are critical parameters in assessing structural integrity through interfacial bonding and were positively affected by the introduction of CNFs. The response of composites to LVI demonstrates the potential of CNFs to increase impact strength by reducing the energy absorbed and the size of the damage introduced. Epoxy matrices reinforced with CNFs showed an average increase in stiffness of 15% and 20% for bending and tensile, respectively. The laminates, on the other hand, showed an increase in bending stiffness of 20% and 15% for tensile and modulus, respectively. In the case of ILSS and ILFT modes I and II, the addition of CNFs promoted average increases in the order of 50%, 100%, and 50%, respectively.

## 1. Introduction

Composite materials offer numerous advantages over traditional materials such as metals or polymers. They are lightweight and offer improved mechanical performance, increased specific strength and fatigue (durability). They contribute to energy efficiency through weight reduction, excellent energy absorption, and thermal insulation. They have high thermal stability, low thermal expansion, and efficient heat dissipation. Additionally, they can be adapted to specific chemical properties to obtain improved properties, such as resistance to environmental degradation, and offer versatility in applications, with fewer parts, manufacturing techniques, and design flexibility. These properties make composites highly beneficial in a wide range of industries [1,2].

In a composite, the matrix acts as a binder that holds the reinforcement together and transfers loads between the reinforcement components. In the case of polymeric composites, epoxy matrices are a versatile class characterized by the presence of two or more oxirane rings or epoxy groups within their molecular structure [3,4]. They exhibit excellent properties, such as strong adhesion to a wide range of substrates, favorable mechanical properties (high strength, temperature resistance, durability, and resistance to extreme environmental conditions), dimensional stability, and design versatility, making them highly versatile for applications in structural materials and coatings. Despite these highly regarded properties, epoxy matrices have some limitations, such as very high viscosity, moisture absorption, low toughness, flammability, intrinsic brittleness after curing, poor fatigue strength, and low fracture energy, especially in high-end applications where better impact resistance and toughness are required. Another disadvantage of epoxy thermosets is that their stiffness and strength decrease significantly in the region of the glass transition temperature [5,6,7]. Their hardness and brittleness render them susceptible to cracking, which compromises their utility as structural adhesives, an area where epoxy matrices find significant application [8].

Epoxy matrices generally have higher modulus and strength than thermoplastics such as polyethylene (PE), polypropylene (PP), and polycarbonate (PC). This makes them more suitable for high-load applications. They maintain their mechanical properties at higher temperatures (120 °C) than most thermoplastics, which tend to soften and lose strength with heat [9]. They typically have higher surface hardness and abrasion resistance than thermoplastics, contributing to their durability in demanding applications. For example, epoxy matrices exhibit superior creep resistance compared to thermoplastics, maintain their shape and structural integrity over long periods of continuous loading, and are less susceptible to environmental stress cracking and UV degradation than many thermoplastics. Epoxies are widely used as adhesives because of their excellent bond strength to a wide variety of substrates, which is generally superior to thermoplastic adhesives. In terms of cost, epoxy matrices are about four times the price of polyester matrices and about twice the price of vinyl ester matrices. In summary, Table 1 compares the epoxy matrix with other common thermosetting and thermoplastic matrices used in the manufacture of composites for various performance criteria.

Based on these weaknesses, in recent years, numerous efforts were made to maximize the properties of polymers by incorporating specific nano-sized fillers [11]. This approach provides a viable alternative, resulting in materials with improved mechanical properties, including tensile and bending strength, compression strength, mode I and mode II interlaminar fracture toughness, and impact properties. One of the most successful strategies for enhancing the chemistry, mechanics, and composition of epoxy matrices is the incorporation of nanofillers into their nanostructures [8,11]. However, the use of nanofillers as performance enhancers is a delicate process, and factors such as the resin-to-filler ratio and the uniformity of filler dispersion within the epoxy matrices are critical determinants of the final properties of the composites.

The dispersion of nanofillers in epoxy resin is critical to achieving the desired improvements in mechanical and other properties. Common techniques for dispersing nano fillers in epoxy resins include mechanical agitation: use of high-speed stirrers to break up agglomerates and disperse nanofillers throughout the resin; ultrasonication: using ultrasonic waves to disperse nanofillers by breaking up agglomerates through cavitation; three-roll milling: passing the nanofiller/epoxy mixture through three rollers to achieve uniform dispersion; and solvent mixing: dissolving both the epoxy and nanofillers in a common solvent, followed by solvent evaporation to achieve a uniform mixture.

On the other hand, there are several mechanisms that affect epoxy performance, being (i) stress transfer: nanofillers with high aspect ratios, such as carbon nanotubes (CNTs) and graphene (GP), can efficiently transfer stress from the epoxy matrix to the nanofillers, significantly increasing the mechanical strength and stiffness of the composite; (ii) crack bridging and deflection: Nanofillers can act as barriers to crack propagation. They can bridge cracks and increase the energy required for crack growth, thereby improving the fracture toughness of the epoxy resin; (iii) interfacial interactions: Strong interfacial bonding between nanofillers and the epoxy matrix is essential for effective load transfer. Functionalization of nanofillers can enhance these interactions, resulting in improved mechanical properties; (iv) stress distribution: nanofillers can improve the stress distribution within the epoxy matrix, reducing stress concentrations and improving the overall strength and durability of the composite; (v) thermal stability: Nanofillers can improve the thermal stability of epoxy resins, allowing them to maintain their mechanical properties at higher temperatures. This is particularly important for applications involving thermal cycling or high temperature environments; (vi) electrical and thermal conductivity: certain nanofillers, such as graphene and CNTs, can impart electrical and thermal conductivity to the epoxy resin, expanding its range of applications in electronics and thermal management; and (vii) viscosity modification: The addition of nanofillers can change the viscosity of the epoxy resin, which can be beneficial or detrimental depending on the application. Controlling filler dispersion and loading is key to maintaining processability [12,13,14,15,16,17].

The influence of carbon nanofillers on the mechanical properties of composites is multifaceted and can be attributed to several mechanisms. Allotropes of carbon improve the mechanical properties of composites through the following primary mechanisms: (i) load transfer: as they have high intrinsic strength and stiffness. When dispersed in a matrix, the load can be effectively transferred from the matrix to the nanofillers, thereby increasing the overall mechanical strength and stiffness of the composite. This load transfer is facilitated by strong interfacial bonding between the nanofillers and the matrix; (ii) stress distribution: Their high aspect ratio and large surface area allow them to act as stress concentrators, redistributing applied stresses more uniformly throughout the matrix. This leads to enhanced resistance to crack initiation and propagation; (iii) crack bridging and deflection: As a crack propagates, carbon nanofillers can bridge the crack by holding the two crack faces together, thereby increasing the energy required for crack propagation. In addition, nanofillers can cause crack deflection, increasing the crack path and hence the toughness of the composite; (iv) interfacial interactions: Strong interfacial interactions between the nanofillers and the matrix are critical. Covalent bonding, van der Waals forces, and mechanical interactions between the matrix and nanofillers can improve the efficiency of charge transfer. Functionalization of carbon nanofillers can enhance these interfacial interactions; (v) alignment and distribution of nanofillers: Proper alignment along the loading direction can maximize reinforcement efficiency. Homogeneous distribution prevents agglomeration and ensures that the nanofillers can interact effectively with the matrix; (vi) matrix reinforcement: Nanofillers can also affect the intrinsic properties of the matrix. For example, they can increase the modulus and strength of the matrix by introducing physical barriers to plastic deformation; and (vii) thermal stability and degradation resistance: Carbon nanofillers can improve the thermal stability of composites, which in turn can improve mechanical properties at elevated temperatures. They can also inhibit matrix degradation, thereby extending the mechanical performance of the composite over time [18,19,20,21].

For this purpose, for example, carbon-based nano-reinforcements garnered significant importance and interest within the scientific community due to their remarkable properties and diverse potential applications in various technologies [22]. They can be found in morphologies such as hollow tubes, ellipsoids or spheres, i.e., carbon nanofibers (CNFs), CNTs, carbon nano-onions (CNOs), carbon nanohorns (CNHs), graphene nanoplatelets (GNPs), fullerenes (C60), carbon black, buckyballs (BBs), mesoporous carbons (MCs), GP, and graphite [22,23,24,25,26].

Comparing the carbon-based nano-reinforcements, CNFs have the highest surface area ranging from 20 to 300 m^2^/g, while CNTs, GP, graphite, and carbon black have surface areas of 50 to 1315, 500 to 2630, 1 to 132, and 10 to 1443 m^2^/g, respectively. Diameter and length are crucial dimensions affecting variables such as dispersion, agglomeration, and property enhancements. CNFs exhibit the highest values with diameters of 10 to 500 nm and lengths of 0.5 to 200 μm, while CNTs have intermediate values (1 to 10 nm diameter, 1 to 100 μm length), and GP has the lowest values (6 to 9 nm diameter). In terms of density, GP has the highest values, followed by graphite, CNFs, CNTs, and carbon black (2 to 2.3, 1.9 to 2.3, 1.5 to 2.0, 0.8 to 1.8, and 0.13 to 2 g/cm, respectively). In general, CNFs have slightly lower electrical conductivities compared to CNTs and GP, with values ranging from 10^−7^ to 103, 102 to 106, and 106 S/cm, respectively. CNTs exhibit lower thermal conductivities compared to most carbon nanoallotropes, followed by GP and CNFs, with values ranging from 2000 to 6000, 600 to 5000, and 5 to 1600 W/(m × K), respectively. Regarding surface area, CNFs have higher values followed by GP, carbon black, CNTs, and graphite, with ranges of 20 to 2500, 500 to 2630, 10 to 1443, 50 to 1315, and 1 to 20 m^2^/g, respectively [24,26].

The selection of the appropriate dispersion process in the manufacture of epoxy matrix nanocomposites is critical to the performance of the resulting composites. The challenge is to create sufficiently strong chemical bonds between the nanofiller and the matrix without causing significant damage to the mechanical properties of the nanofiller.

For epoxy matrix nanocomposites, the combination of high-performance dispersion methods, such as high shear mixing, and simultaneous application of an ultrasonic bath is the simplest and most convenient approach to improve the dispersion of nanosized fillers in an epoxy matrix. In this method, the nanofillers are first dispersed in the resin or dissolved in certain solvents, followed by dispersion based on the viscosity of the epoxy resin. Shear mixing can be achieved by magnetic or mechanical stirring. Higher shear forces are required to generate greater shear energies, resulting in more effective dispersion of the nanofillers. 

While the primary focus of this paper is on the performance enhancement of epoxy resins through the addition of nanofillers, in particular CNFs, it is important to recognize the cost implications associated with these advanced materials. The cost of nanofillers is a significant factor in their technical application and commercialization. Different types of nanofillers have different cost structures due to their raw materials, production methods, and market availability: CNTs are one of the most effective nanofillers for improving mechanical and electrical properties, but their production costs are relatively high, especially for high-purity, single-wall CNTs; GP offers excellent mechanical and conductive properties, but is also expensive due to complex production processes such as chemical vapor deposition or exfoliation; CNFs are generally less expensive than CNTs and GP, offering a good balance between cost and performance; nanoclays are relatively less expensive than carbon allotropes and offer good mechanical and barrier properties; and silica nanoparticles, alumina nanoparticles, and other inorganic nanofillers tend to be less expensive but may offer different property enhancements [27].

Incorporating nanofillers into epoxy resins can affect overall production costs due to material costs. The direct cost of nanofillers is a significant component; for example, high-quality, functionalized nanofillers can be particularly expensive. Achieving uniform dispersion of nanofillers may require additional processing steps, such as ultrasonication, three-roll milling, or solvent mixing, which can increase manufacturing complexity and cost. Large-scale production can benefit from economies of scale, reducing the unit cost of nanofiller-enhanced epoxy composites. However, the initial investment in equipment and process development can be significant.

When considering the use of nanofillers, a cost–benefit analysis should be performed to evaluate the trade-offs between increased cost and improved properties of the final composite. The significant improvements in mechanical, thermal, and electrical properties provided by nanofillers may justify the higher material and processing costs, especially for high-performance applications as in the aerospace, automotive, defense industries, biomedical, electronics, and energy sector. Improved durability, thermal stability, and resistance to environmental degradation can lead to longer service life and reduced maintenance costs, providing long-term economic benefits [28].

There are several ways to potentially reduce the cost of nanofillers and make them more viable for widespread use. Improvements in the synthesis and functionalization processes for nanofillers can reduce their cost. For example, scalable methods to produce graphene and CNTs at lower cost are under active research. Efficient functionalization methods can improve the compatibility of nanofillers with epoxy resins, potentially reducing the amount needed to achieve desired properties. Combining different types of nanofillers or using a blend of nano- and microfillers can balance cost and performance [29].

## 2. Carbon Nanofibers

CNFs exhibit exceptional properties, such as a high aspect ratio and molecular orientation, large specific surface area, small pore size, minimal defects, low density, high specific modulus and strength, excellent electrical and thermal conductivity, remarkable chemical resistance, high temperature tolerance, low thermal expansion, and outstanding mechanical performance. These remarkable attributes render CNFs highly desirable for a wide range of applications [30,31,32].

Incorporating CNFs into nanocomposites offers several potential benefits, including the increase in mechanical properties without changing the mass. These improvements encompass various aspects, such as increased modulus [33,34,35,36], strength [33,34,35,36], fracture toughness [37,38,39], fatigue strength [37,40], delamination resistance and damage tolerance [41], impact strength [42], and structural damping [43]. Furthermore, the addition of CNFs can lead to improved electrical and thermal conductivity, heightened thermal stability, enhanced flame retardancy, superior barrier properties, and reduced susceptibility to environmental factors such as moisture absorption and degradation caused by irradiation [44].

These nano-reinforcements can be described as 1D carbonaceous filaments with nanometer-scale dimensions (ranging from 3 to 100 nm in diameter). They are composed of stacked graphene layers that exhibit a specific orientation with respect to the fiber axis. These materials are typically categorized into three groups based on the angle between the graphene layers and the growth axis: parallel (angle = 0°), fishbone (0° < angle < 90°), and platelet (angle = 90°). The synthesis of CNFs involves various methods, leading to diverse structures such as porous, hollow, helical, twisted, and stacked forms. Achieving these structural variations is possible through instrumental techniques and experimental design [45,46].

CNFs differ from CNTs in that they lack the tubular nanostructure typically found in nanotubes. Instead, CNFs exhibit a fishbone or platelet-shaped arrangement, where graphene planes are stacked to form a 1D fiber morphology, as shown in Figure 1. Similar to other carbon nanofilaments, CNFs are mainly produced by catalytic chemical vapor deposition (CCVD) from a carbon feedstock (light or aromatic hydrocarbons) using an elemental transition metal (Fe, Ni, Co, and Cu) as a catalyst, in a hydrogen atmosphere (partial) at temperatures ranging from 500 to 1200 °C [47]. Therefore, the only difference between the various forms of CNFs is their chemical structure:

(i)Platelet CNFs exhibit a structure where graphene layers are oriented perpendicular to the fiber axis, as shown in Figure 1a. These fibrils typically have a width of around 100 nm, and the presence of hydrogen or other heteroatoms is necessary for stabilizing the plates [47]. In the case of bidirectional fibers, a solid particle is usually located in the middle of the fiber [50]. On the other hand, ribbon CNFs feature a stacked arrangement of graphene layers parallel to the fiber axis, as depicted in Figure 1b. Additionally, CNFs can also exhibit a coiled configuration, as illustrated in Figure 1f.(ii)Fishbone CNFs are characterized by the inclination of graphene layers in relation to the fibril axis. The presence of hydrogen is necessary to stabilize the edges of these CNFs. There are two variations of fishbone CNFs: those with a hollow core, as shown in Figure 1d, and those with a solid core, as depicted in Figure 1c [51].(iii)Ribbon CNFs consist of unrolled graphene layers that are straight and parallel to the fibril axis. They have non-cylindrical cross-sections, as shown in Figure 1e. Regarding the position of the catalytic solid particle, there is agreement among researchers that it is located at one extreme. However, there is some discrepancy among authors regarding the orientation of the graphite layers in relation to the fibril axis. Some claim that the layers are completely parallel [50], while others suggest that they are slightly inclined [52].(iv)Stacked-cup CNFs exhibit a continuous layer of rolled (spiral) graphene along the fiber axis. This spiral orientation results in a truncated cone arrangement along the axis, with a wide internal hollow space, as shown in Figure 1g.

In the production of different configurations of CNFs, different production techniques are applied according to their properties and applications. These include catalytic chemical vapor deposition (CCVD) [53], electrospinning/electrospun [54], plasma-enhanced chemical vapor deposition (PECVD) [55,56], gas-phase flow catalytic method [57], templating [58], phase separation [59], arc discharge deposition [60], and floating catalyst (FC) [61]. These fabrication techniques offer versatility in controlling the morphology, structure, and properties of CNFs, enabling their application in a wide range of fields.

CCVD is a widely used method for producing vapor-grown carbon nanofibers (VGCNFs). This process, originating in the 1970s, was refined to specifically generate CNFs. In this technique, a catalytic thermal chemical vapor in a quartz tube electric furnace involves the use of C_2_H_2_ as the carbon source. Various metals or alloys, such as iron (Fe), cobalt (Co), nickel (Ni), chromium (Cr), and vanadium (V), act as catalysts to dissolve carbon into metal carbide. Carbon sources such as molybdenum (Mo), methane (CH_4_), carbon monoxide (CO), synthesis gas (H_2_/CO), or ethane (C_2_H_6_) are utilized within a temperature range of 700 to 1200 K. The CCVD procedure includes stages where catalyst powder is heated, carbon sources are introduced, and the temperature is controlled to synthesize CNFs (Figure 2a). Post-processing involves purification steps in nitric acid (HNO_3_) and hydrochloric acid (H₂O:HCl) solutions, followed by washing with distilled water and isopropyl alcohol. However, CCVD has limitations in producing short, challenging-to-align fibers, hampering their use in applications. The growth mechanism of CNFs depends on the catalyst’s surface geometry and the gaseous carbon feedstock during processing. The resulting CNF structures are influenced by manufacturing techniques. The size of catalyst particles typically dictates the CNF’s outer diameter, with VGCNFs having unique structures resembling annular rings, characterized by sp^2^ graphite. Metal particle size affects fiber thickness, while growth temperature and metal type influence fiber orientation. Cup-stacked and platelet CNFs are two distinct types generated by CCVD, each exhibiting specific structures; cup-stacked CNFs have a helically folded graphene layer, forming a hollow core with cup-shaped appearances along the fiber axis. This unique structure differentiates VGCNFs from CNTs, which resemble single or multiple concentric cylinders formed by parallel-oriented graphene layers [45,62,63].

Electrospinning is a versatile technique that can be used to produce CNFs, where a polymer solution containing a carbon precursor, such as polyacrylonitrile (PAN) or a blend of PAN and other polymers, is electrospun. The resulting polymer nanofibers are then subjected to a thermal treatment process, known as carbonization, to convert them into CNFs (Figure 2b). Electrospinning is a widely utilized technique for synthesis of CNFs. This method enables the production of polymeric nanofibers, offering advantages such as ease of control and environmental compatibility. It is considered a flexible and robust method for large-scale production of organic polymer or composite nanofibrous mats, having diameters ranging from submicrons to nanometers, and demonstrating good electrospinnability and stability. The process involves a setup comprising a metallic spinneret, syringe pump, high-voltage power supply, and grounded collector within a controlled environment. A polymer solution is pumped through the spinneret while a high voltage is applied between the spinneret and the collector, resulting in fine filaments that deposit randomly on the collector, forming a nanofiber web. Improved techniques were developed to generate aligned nanofiber arrays, porous surfaces, and large-scale production. Rotating drum collectors enhanced homogeneity in fiber deposition, ensuring uniform thickness. Parameters such as polymer solution type, solvent, capillary size, flow rate, working distance, and applied potential significantly impact CNF properties. A subsequent heat treatment carbonizes the polymer nanofibers to form CNFs. Solution concentration, viscosity, and temperature also influence fiber dimensions, with higher temperatures potentially inducing beta-phases and increased diameter correlating with solution concentration increment following a power law relationship [45,46,62,63].

PECVD is a technique in which plasma is utilized to enhance the deposition of thin films or nanostructures onto a substrate. In the case of CNFs, a carbon-containing precursor gas, such as methane or acetylene, is introduced into a PECVD chamber. The plasma dissociates the precursor gas, and the carbon species deposit and grow into CNFs on the substrate (Figure 2c). PECVD is an efficient alternative method for low-temperature production of CNFs as it provides improved control over synthesis parameters such as plasma power, reaction temperature, gas species, and carbon source. Experimental results show that nanostructures produced via PECVD exhibit better alignment compared to nanostructures synthesized using thermal processes. PECVD-based methods offer advantages for CNF fabrication, including control over nanofiber diameter, length, and alignment, enabling the growth of individual, freestanding, and vertical carbon nanostructures. In contrast, thermal CVD typically yields either spaghetti-like films or ensembles resembling towers. Upon closer inspection, the individual nanostructures within the tower are found to be wavy [56,68].

In the gas-phase flow catalytic method, the catalyst precursor undergoes direct heating, followed by introduction into the reaction chamber alongside the hydrocarbon gas. This leads to decomposition of the hydrocarbon gas and the catalyst at two distinct temperature zones (Figure 2d). Consequently, the decomposed catalyst aggregates into nanoscale particles. Subsequently, CFs are synthesized utilizing these nanoscale catalyst particles [69]. Lastly, the decomposed catalyst particles resulting from the organic compound can be dispersed throughout a 3D space. This method enables straightforward management of volatilization quantities. Consequently, it allows for substantial production of CF in a shorter duration, facilitating uninterrupted carbon fiber fabrication [70].

Templating provides precise control over the size, shape, and distribution of carbon nanofibers, making it a versatile and powerful technique for their production. The hard template method uses solid templates, such as anodic aluminum oxide (AAO) or mesoporous silica, while the soft template method uses self-assembling structures, such as surfactant micelles or block copolymers. Both methods involve infiltration with a carbon precursor, followed by polymerization, carbonization, and template removal to produce high-quality CNFs (Figure 2e) [58,65].

In addition to these more industrial techniques, others were suggested in the literature, and duly accepted by the scientific community, with the aim of improving their physical and/or mechanical characteristics in order to obtain better adhesion to the matrix [63]. For example, phase separation is a technique involving dissolution, gelation, and extraction using varied solvents, freezing, and drying methods, leading to the creation of a nanoporous foam (Figure 2f). However, the entire transformation from solid polymer to nanoporous foam is time-consuming. Self-assembly, involving the spontaneous arrangement of dispersed components into an organized structure, occurs through local interactions among the components. This technique, akin to phase separation, is also time-consuming, especially in the continuous production of polymer nanofibers [70].

Arc discharge deposition is a method in which a graphite rod acts as the negative cathode and a positive anode when placed a few millimeters apart (Figure 2g). By applying a high current, the graphite rod is evaporated, resulting in the formation of carbon products that deposit on the chamber walls or the cathode substrate. Both CNTs and CNFs can be synthesized using arc discharge alternating current (AC) or direct current (DC) [71].

The FC method involves the production of VGCNFs through the catalytic decomposition of a hydrocarbon, such as benzene or methane, in a hydrogen atmosphere within a temperature range of 700 to 1200 °C (Figure 2h). During the FC method, the nanofibers float in the reactor space [61]. This method offers several advantages, including a relatively low cost, scalability, and the ability to produce large quantities of CNFs, making it suitable for industrial-scale implementation. However, achieving the desired properties of the nanofibers requires careful control of process parameters such as catalyst concentration, gas composition, temperature, and residence time. Insufficient contact time between catalyst particles and carbon sources can result in incomplete transformation of hydrocarbons into graphitic products [72].

Finally, Table 2 summarizes the main properties of CNFs produced by some processes available in the literature.

## 3. Effect of CNFs on the Static Response of Epoxy Matrix Composites

### 3.1. Static Properties of CNF Multiscale Epoxy Matrix and Composites

It is well described in the literature that when an epoxy matrix is reinforced with CNFs, its mechanical performance can be improved significantly. However, these improvements depend on several factors, such as the aspect ratio of the CNFs, the dispersion and distribution quality, alignment, the adhesion and interface between the reinforcement and matrix, the formation of agglomerates, voids between the CNFs’ surfaces and the matrix due to the weak adhesion of the fiber to the matrix, and processing techniques. In the specific case of laminated composite materials, their main advantage consists of delaying delamination (Figure 3). Therefore, the main results obtained with a view to maximizing the mechanical properties of composite materials are summarized below, with a special focus in this section on the static response in bending and tensile modes.

In this context, Table 3 summarizes the studies carried out on the effect of the CNF content on the bending response of epoxy resins, while Figure 4 quantifies the improvements achieved. For this purpose, the abscissa axis represents the percentage by weight of CNFs (wt.%), on a logarithmic scale, and on the ordinate axis the percentage improvement in bending strength and modulus. Figure 4a shows the increase in bending strength achieved by incorporating CNFs into the epoxy matrix. The studies show varying degrees of improvement, with some studies showing significant improvements for specific CNFs loads. Similarly, the Figure 4b shows the percentage increase in bending modulus due to CNF reinforcement. The results evidence that the bending modulus is also improved with the addition of CNFs.

This analysis shows that it is possible to maximize the static properties of epoxy resins, regardless of their mechanical, physical, and chemical properties, by adding an optimum percentage by weight of CNFs. Patton et al. [80], for example, carried out one of the first studies in this field and observed improvements of around 97.2% in the bending modulus and 36.7% in the bending strength of an epoxy matrix through the incorporation of 18.2 wt.% CNFs. The authors used an acetone/epoxy solution infusion method through a CNF mat under vacuum to prepare the nanocomposites. Iwahori et al. [81], used cup-stacked carbon nanofibers (CSNFs), which were dispersed into an epoxy resin in two different weight percentages (5.0 and 10.0 wt.%) and with two CSNFs aspect ratios of 10 and 50. They observed that increasing the weight content of CSNFs resulted in higher bending strength and modulus values, achieving increases in the order of 38% and 42% higher when 10.0 wt.% CSNFs were used, respectively. Pervin et al. [82] introduced a novel technique for fabricating nanocomposite CNFs/epoxy by combining ultrasonic cavitation using a high-intensity ultrasonic liquid processor and high-speed mechanical agitation. The results of bending tests demonstrate an enhancement in both modulus and strength as the loading percentage of CNFs increased. Specifically, the bending modulus and strength exhibited a progressive increase with higher weight percentages of CNFs. When 4.0 wt.% of CNFs was added, there was a 27% improvement in bending modulus and a 17% improvement in strength. This highlights the positive impact of incorporating CNFs into the epoxy matrix using their developed technique. Zamu et al. [83] produced nanocomposites with herringbone graphitic nanofibers (GNFs). Loadings of 0.15, 0.2, 0.3, 0.5, and 1.3 wt.% GNFs are compared with neat epoxy, and the highest mechanical properties were achieved by the nanocomposites containing 0.3 wt.% GNFs. At this filler amount, the bending strength, modulus, and breaking strain of the nanocomposite are increased by about 25.9, 20.6, and 30.8%, respectively. Bal [84] fabricated epoxy nanocomposites of different contents of CNFs. The bending modulus increases 33%, 60%, and 49% for composites with 0.5, 0.75, and 1.0 wt.% CNFs. As a result of CNFs’ agglomeration at higher concentration, their high aspect ratio, and Van der Waals attractive interactions, the bending property could be reduced. Ardanuy et al. [85] prepared VGCNFs/trifunctional epoxy resin composites, and used different weight fractions of VGCNFs, from 0.05 to 2.0 wt.%. The composite was prepared by directly mixing the VGCNFs and the epoxy resin using ultrasounds to improve dispersion. Compared to the neat epoxy, the maximum enhancement of the bending modulus was found for 0.1 wt.% VGCNFs composites, with 21.2% improvement.

Regarding the benefits of the CNF content on the bending response of epoxy matrix composite laminates, Table 4 provides an overview of the experimental work carried out on this subject, while Figure 5 summarizes the benefits obtained by each researcher reported above. One more time, the abscissa axis represents the percentage by weight of CNFs (wt.%), on a logarithmic scale, and on the ordinate axis, the percentage improvement in bending strength and modulus. Figure 5a shows the increase in bending strength for composite laminates with CNF reinforcement, and Figure 5b illustrates the increase in bending modulus for composite laminates reinforced with CNFs. Different results show different degrees of improvement in mechanical properties, for example, the bending modulus improves significantly with the incorporation of CNFs.

Zhou et al. [86,87] analyzed the effect of the weight content of CNFs, which were infused into the epoxy resin using high-intensity ultrasonic irradiation. They observed agglomerates of CNFs for fractions above 2.0 wt.%, while for lower values, the methodology used proved to be very suitable. Furthermore, the introduction of 2.0% by weight of CNFs led to a 23.3% improvement in bending strength and a 1% improvement in bending modulus. Li et al. [75] found that the addition of 20 g/m^2^ of CNFs (approx. 12.7 wt.% CNFs) to carbon composites, prepared using a powder method for dispersing the nanofiller in the middle plane by hand lay-up of the laminate and manufactured by the autoclave process, resulted in an approximate increase of 7.1% in bending strength and 10.1% in bending modulus compared to the control carbon composite. Green et al. [88], by adding 0.1 wt.% and 1.0 wt.% of CNFs to produce multiscale GF, reinforced composites applying the VARIM process, the bending strength increased by 17% and 20%, respectively, and the bending modulus increased 23% and 26%, respectively, when compared with the control composite. In the work of Miranda et al. [89], CNFs were grown in situ over the surface of a CF fabric by CVD, followed by laminate manufacturing by hot pressing. The results show an overall reduction in mechanical properties as a function of added CNFs, for 1.0 wt.% CNFs, the bending strength increased to 17%, and the bending modulus had no influence on the deposition of CNFs over the surface of CF fabrics.

Chen et al. [90] used in their study small mass fractions (i.e., 0.1%, 0.3%, and 0.5%) of (expanded carbon nanofibers) ECNFs, VGCNFs, and GCNFs surface-functionalized and oxidized with nitric acid and hexanediamine for the fabrication of hybrid multi-scale composites with woven fabrics of CF via the technique of vacuum-assisted resin transfer molding (VARTM) and compared the mechanical properties. The results indicate that the incorporation of 0.5% ECNFs in the epoxy resin resulted in the improvements of bending strength by 13%. In general, the reinforcement effect of ECNFs was similar to that of VGCNFs, while it was higher than that of GCNFs, as indicated in Table 3. The results of mechanical properties of hybrids multi-scale carbon fiber-reinforced polymers (CFRP) allow us to conclude that the optimal mass fraction of ECNFs, VGCNFs, and GCNFs in the nano-epoxy resin is 0.3%. For example, the nano-epoxy resin with 0.3% VGCNFs resulted in a bending strength of 18.3%. Singer et al. [91] produced CFRP specimens that were tested via a four-point bending test. The mechanical properties, such as bending modulus and bending strength, were improved by adding 0.7 wt.% CNFs to the resin compared to neat epoxy in 14%.

**Table 3 polymers-16-02152-t003:** Summary of studies related to the effect of CNFs on the bending properties of epoxy matrix nanocomposites.

Autor, Ref.	CNFs Type	CNFs Integration Method	Optimum Loading (CNFs wt.%)	Bending Strength [MPa] (Increase (%))	Bending Modulus [GPa] (Increase (%))	Failure Strain [%] (Increase (%))
Patton et al., [80]	VGCNFs	Acetone/epoxy solution infusion.	18.2	123.0 (36.7%)	7.85 (97.2%)	-
		High shear mixing.	15.5	112.5 (29.3%)	6.18 (169.9%)	-
		Blender and two roll mill mixing.	19.2	146.4 (67.9%)	7.02	-
Xu et al., [92]	GCNFs	Mixed and sonicated.	0.3	139.6 ± 4.05 (28.25)	3.07 (0.52%)	3.93 (36.7%)
Iwahori et al., [81]	CSNFs	Mechanical mixing, passing through a vacuum chamber and post-cure in a hot press.	10.0	135.8 (37.7%)	3.277 (41.9%)	-
Pervin et al., [82]	CNFs	Ultrasonic cavitation.	4.0	99.4 ± 4.6 (17%)	2.81 ± 0.12 (27%)	-
Zhamu et al., [83]	GNFs	Mixed by low-power sonication and cured in a vacuum oven.	0.3	166.4 ± 2.0 (25.9%)	3.356 ± 0.056 (20.6%)	0.068 ± 0.005 (30.8%)
Sui et al., [93]	CNFs	Mechanical mixing.	0.3	137.7 ± 4.2 (32%)	2.92 ± 0.04 (9%)	7.5 ± 1.9 (70%)
Sancaktar et al., [94]	ECNFs	Non-woven ECNFs fabrics were impregnated with epoxy resin.	0.98	~220.0 (−33%)	-	-
Bal, [84]	CNFs	Dispersed in acetone by sonication, mixed with resin and sonicated at controlled power levels, and followed degassing process in vacuum oven.	0.75	-	2.69 (60%)	-
Ardanuy et al., [85]	VGCNFs	Mixing by hand and ultrasound bath.	0.1	105.0 ± 15 (4%)	4.0 ± 0.3 (21.2%)	3.8 ± 1.0 (−34.2%)
Zhang et al., [95]	VGCNFs	Ultrasonic and then mixing followed by ultrasonic again.	0.2	~120.0 (over 200%)	~2.7 (under 10%)	-
Zhang et al., [96]	CNFs	Ultrasonically dispersed, mixed, and rotary evaporator.	0.3	~80.0 (over 400%)	-	-
Shokrieh et al., [97]	VGCNFs	High speed mechanical mixing and sonicated via probe sonicator.	0.25	~121.0 (10%)	~3.18 (6%)	-
Chen et al., [90]	ECNFs	Surfaces oxidation and functionalization. The nano-epoxy mixture was first subjected to ultrasonication, followed by mechanical stirring and degassing and finally post-curing.	0.5	412.3 (10%)	18.8 (14.6%)	-
	VGCNFs			424.6 (13.3%)	18.2 (11.0%)	-
	GCNFs			418.7 (11.7%)	18.5 (12.8%)	-
Ahmadi et al., [98]	CNFs	Dispersed in acetone/epoxy resin under mechanical stirring by high-speed, sonicated, and vacuum oven.	1.0	213.6 ± 4.4 (97.8%)	5.14 ± 0.28 (143.6%)	-
Zeltmann et al., [99]	CNFs	Dispersing was accomplished using a mechanical mixer with a high shear impeller, and cured at RT and post-cured at 90 °C.	1.0	96.9 (−8.6%)	~2.3 (~5%)	-
Gao et al., [33]	CNFs	Rigorously agitation.	3.0	163.9 ± 7.8 (49.2%)	6.2 ± 0.4 (82.4%)	-
Danni et al., [54]	CNFs	Dissolution, magnetic stirring, and sonication to obtain CNF mats. Hand lay-up method to manufacture de composite.	3.0	122.58 (97.2%)	-	-
Nimbagal et al., [42]	CNFs	EP was preheated, mixed manually, and sonication and cured at RT.	0.3	76.9 (48.74%)	-	-
Santos et al., [34]	CNFs	Simultaneous dispersion in a high-speed shear mixer at high shear rate and sonication for 3 h at RT followed by degassing.	0.75	118.7 ± 1.2 (11.7%)	3.0 ± 0.08 (11.7%)	5.2 ± 0.2 (−2%)
			0.5	123.4 ± 2.8 (11.7%)	3.2 ± 0.09 (11.5%)	5.4 ± 0.5 (−6.3%)

**Table 4 polymers-16-02152-t004:** Summary of studies related to the effect of CNFs on the bending properties of composite laminates.

Autor, Ref.	Fiber/Matrix	CNFs Type	CNFs Integration Method	Manufacture Process	Optimum Loading (CNFS wt.%)		Bending Strength [MPa] (Increase (%))	Bending Modulus [GPa] (Increase (%))	Failure Strain [%] (Increase (%))
Iwahori et al., [81]	CF/EP	CSNFs	Mechanical mixing, passing through a vacuum chamber, and post-cure in a hot press.	Hand lay-up, vacuum application, and post-cure in a hot press.	10.0		789.5 (18.3%)	53.5 (4%)	-
Zhou et al., [86]	CF/EP	CNFs	High-intensity ultrasonic processing followed by high-speed mechanical mixing and high vacuum.	VARTM	2.0		597.0 ± 21 (22.3%)	49.4 ± 3.1 (1%)	1.27 ± 0.03 (8.5%)
Li et al., [75]	CF PrP/EP	VGCNFs	Powder method (applied in the middle plane by hand lay-up process).	Autoclave	12.7		1283.7 (7.1%)	114.1 (10.1%)	-
Green et al., [88]	E-GF/EP	CNFs	Mechanical mixer.	VARIM and compress	1.0		404.0 ± 18.6 (20%)	22.0 ± 0.5 (26%)	-
Bortz et al., [100]	CF/EP	CNFs	Hand mixing and TRM.	VARTM	1.0		~310 (over 9%)	~11.3 (over 10%)	-
Chen et al., [101]	CF/EP	ECNFs	Fabrication of mats of ECNFs.	VARTM with interlaminar regions containing mats of ECNFs.	~2.5		418.5 ± 11.7 (11%)	32.8 ± 7.8 (9%)	-
Miranda et al., [89]	CF/EP	CNFs	CNFs grown onto the surface of carbon fiber fabrics.	Hot pressed	1.0		~380.0 (17%)	31.5	-
Ali et al., [102]	CF/EP	CNFs	CNFs dispersed using a high shear mix, sonicated in a bath ultrasonicator followed by a spray-up process.	VARTM	1.0		~160.0 (over 40%)	~30.0 (19%)	-
Chen et al., [103]	CF/EP	ECNFs	Thermal treatments of stabilization in air followed by carbonization in argon.	VARTM	14.0 (Collection time at 10 min)		465.6 ± 38.4 (23.5%)	24.8 ± 3.9 (105%)	-
Dhakate et al., [35]	CF PrP/EP	CNFs	Mixed and sonicated.	Impregnated and was applied temperature and pressure (hot plate).	1.1		730.0 (83.4%)	~40.0 (over 100%)	-
Chen et al., [90]	CF/EP	ECNFs	Surfaces oxidation and functionalization. The nano-epoxy mixture was first subjected to ultrasonication, followed by mechanical stirring and degassing and finally post-curing.	VARTM	0.3		545.0 ± 9.5 (13.6%)	-	-
		VGCNFs					567.3 ± 21.8 (18.3%)	-	-
		GCNFs					552.6 ± 44.8 (15.2%)	-	-
Jie. et al., [104]	CF/EP	CNFs	CVD	Manual stacking followed by heating at 2300 °C.	1.0	CF parallel	-	187.92 (45.6%)	-
						CF vertical	-	11.23 (56.6%)	-
Singer et al., [91]	CF/EP	CNFs	TRM dispersion.	Infiltration and cure in a hot press.	0.7		~600.0 (14%)	~47.5 (14%)	-
Kattaguri et al., [105]	GF/EP	CNFs	EP resin was preheated, mechanical mixing at high speed, sonication, and degassing.	Hand lay-up followed by hot pressing.	1.0		~415.0 (29.0%)	~23.5 (~7%)	-
De et al., [106]	CF/EP	CNFs	Electrophoretic deposition (EPD) technique.	Hand lay-up, followed hot pressing.	0.5 [g/L]		191.0 ± 7.81 (6.7%)	13.0 ± 0.87 (30%)	3.14 ± 0.2 (36%)
Kar et al., [107]	GF/EP	CNFs	Magnetic stirring dispersion, ultrasonication, and degassing.	Hand lay-up, followed hot pressing.	1.0		~380 (~13%)	~20.0 (~8%)	-
Santos et al., [36]	CF/EP	CNFs	Simultaneous dispersion in a high-speed shear mixer at high shear rate and sonication for 3 h at RT, followed by degassing.	Hand lay-up and simultaneous application of vacuum and pressure.	0.75		905.3 ± 13.9 (20.4%)	61.4 ± 1.8 (11.4%)	1.5 ± 0.08 (−7%)
					0.5		850.9 ± 46.6 (12.5%)	51.7 ± 1.5 (8.8%)	1.8 ± 0.09 (4%)

**Figure 4 polymers-16-02152-f004:**
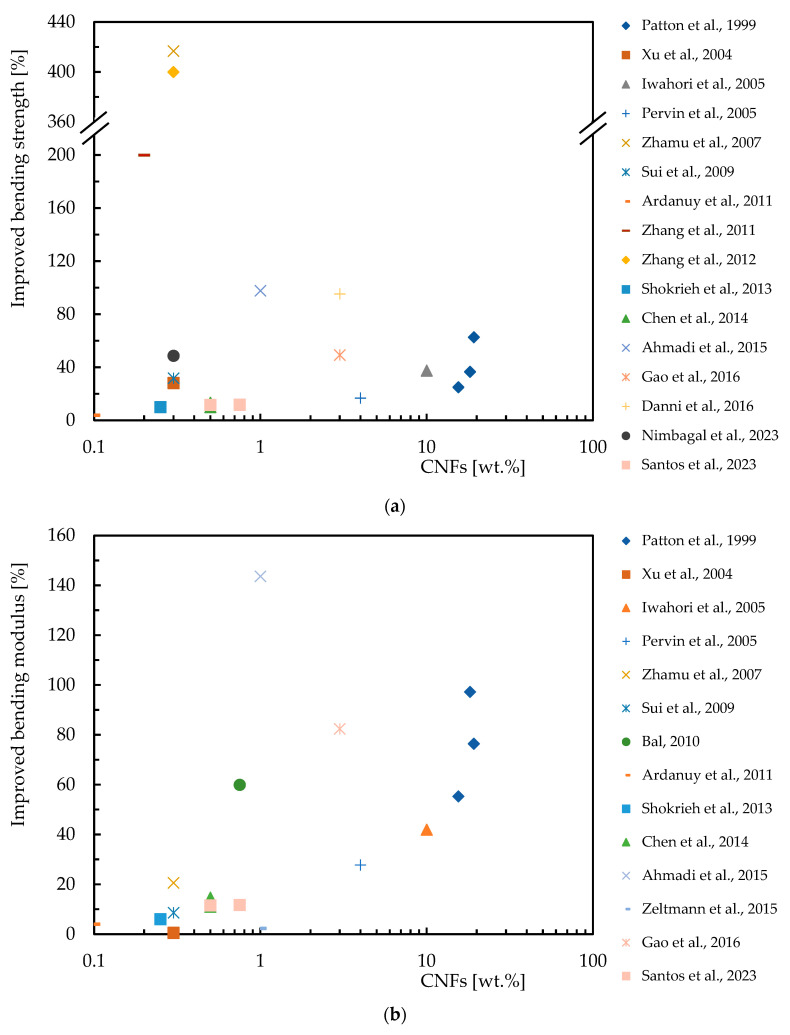
Benefits achieved by epoxy matrix nanocomposites with CNFs in terms of: (**a**) Bending strength; (**b**) Bending modulus. Data obtained from [33,34,42,59,80,81,82,83,84,85,90,92,93,94,95,96,97,98,99].

**Figure 5 polymers-16-02152-f005:**
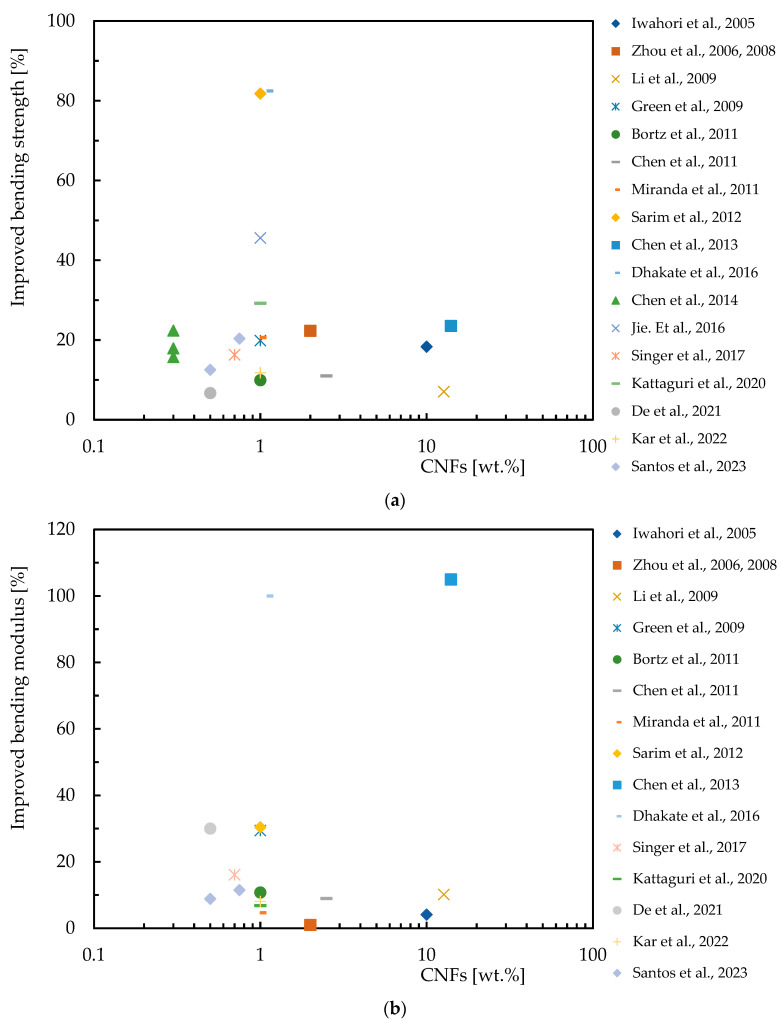
Benefits achieved by composite laminates with CNFs in terms of: (**a**) Bending strength; (**b**) Bending modulus. Data obtained from [35,36,75,81,86,88,89,90,91,100,101,102,103,104,105,106,107].

One of the most crucial factors contributing to the enhancement of both tensile strength and tensile modulus in epoxy nanocomposites is the effective dispersion of the CNFs [108,109,110], because aggregations/agglomerations promote non-uniform stress distribution with consequent decrease in the tensile properties [111,112]. Therefore, when CNFs are homogeneously dispersed, they form a continuous network within the epoxy matrix, leading to a more efficient transfer of the load in the epoxy matrix, and consequently, an improvement in tensile strength. In addition, this network can effectively restrict the movement of the polymer chains with a consequent increase in tensile modulus [111]. 

There are several studies in the literature aimed at maximizing the tensile properties of nanocomposites, which are summarized in Table 5 for epoxy matrices reinforced with CNFs, while Figure 6 shows the improvements obtained by the various authors. In this representation, the *x*-axis shows the percentage by weight of CNFs (in a logarithmic scale), and the *y*-axis the percentage increase in tensile strength and tensile modulus. Figure 5b and Figure 6a show the results of different studies that highlight the improvements in tensile properties for epoxy matrix nanocomposites. The results evidence that even small additions of CNFs can significantly improve the tensile properties, but improvements depend on factors such as the CNF dispersion method used.

Choi et al. [113] found that CNFs/epoxy nanocomposites showed maximum tensile strength and modulus with 5.0 wt.% CNFs, along with a lower fracture strain with increasing filler content. Similarly, Zhou et al. [40,87,114] observed that the modulus of CNFs/epoxy nanocomposites increased continuously with increasing CNF content, but the tensile strength decreased with further increasing CNF content beyond 2.0 wt.%. Improvements of 17.3% and 14% were achieved for tensile strength and tensile modulus, respectively. Rana et al. [109,110] uniformly dispersed VGCNFs in an epoxy matrix using ultrasonic treatment assisted by high-speed mechanical stirring. This dispersion technique resulted in a significant improvement in tensile strength and modulus, where only 0.5 wt.% of CNFs led to improvements of around 65% and 35%, respectively. In another study, these authors compared the effect of adding original and functionalized CNFs, in 0.1 wt.%, observing that the tensile strength increased by 59.5% and 62.1%, respectively, while the tensile modulus increased by 24.3% and 12.3%, respectively [115]. In a similar study, Liu et al. [116] used 0.4 wt.% CNFs and obtained an increase of 4.7% and 1.9% in tensile strength and modulus with unmodified nanoparticles, respectively, and 12.5% and 6.1% with modified nanoparticles, respectively. Wang et al. [117] obtained improvements of 22.5% in tensile strength and 7.6% in tensile modulus with 0.5 wt.% of CNFs, in addition to improving the durability of the nanocomposites in hygrothermal environments, because CNFs inhibit the hydrolysis of the resin and water absorption.

Concerning the advantages of the tensile response of CNF-reinforced epoxy matrix composite laminates, Table 6 provides a comprehensive overview of experimental studies available in the literature and Figure 7 shows a concise summary of the benefits found. Despite the great interest in this topic, the number of studies available in literature is much smaller than those carried out with epoxy resin. Figure 7 summarizes the benefits of CNF reinforcement in composite laminates in terms of tensile strength (Figure 7a) and tensile modulus (Figure 7b). The results indicate that CNF reinforcement can significantly improve the tensile properties of composite laminates, but similar to nanocomposites, the benefits depend on the quality of the dispersion and the dispersion methods used.

Rana et al. [110] studied composites reinforced with carbon fibers and a matrix nano-enhanced with 0.5 wt.% CNFs and, compared to composites with neat resin, obtained increases in tensile strength and modulus of around 18% and 37%, respectively. Sarim et al. [102] used the spraying technique to reinforce laminates with CNFs (1.0 wt.%), and obtained increases in tensile strength and modulus of around 22.5% and 14%, respectively. Anjabin et al. [118], by functionalizing CNFs and subsequently preparing composites using the hand lay-up technique, achieved a 12.7% improvement in tensile strength and a 32.9% improvement in tensile modulus for 0.3 wt.% CNFs.

**Table 5 polymers-16-02152-t005:** Summary of studies related to the effect of CNFs on the tensile properties of epoxy matrix nanocomposites.

Autor, Year, Ref.	CNFs Type	CNFs Integration Method	Optimum Loading (CNFs wt.%)		Tensile Strength [MPa] (Increase (%))	Tensile Modulus [GPa] (Increase (%))	Tensile Strain [%] (Increase (%))
Ying et al., [119]	CNFs	A homogeneous suspension of surfactant coated CNFs was obtained by magnetic stirring, ultrasonics and high shear stirring. Subsequently, it was mixed and stirred with epoxy resin.	2.0		62.0 (19.2%)	1.385 (17.7%)	8.04 (36.3%)
Iwahori et al., [81]	CSNFs	Mechanical mixing, passing through a vacuum chamber and post-cure in a hot press.	10.0		88.6 (21.6%)	3.602 (45.3%)	3.75 (−40.7%)
Choi et al., [113]	VGCNFs	For low viscosity epoxy, the VGCNFs were dispersed in an acetone by sonication and stirring at RT. The epoxy resin was added to the mixture by sonication and stirring under the same conditions. For high viscosity epoxy, the same procedure as above was followed, but without the acetone treatment. The materials were cured and post-cured at RT.	5.0	Low viscosity	~75.0 (15.4%)	~10.1 (90%)	~1.87 (−25.2%)
				High viscosity	~71.0 (12.7%)	~9.9 (80%)	~1.8 (−28%)
Zhou et al., [40]	VGCNFs	High-intensity ultrasonic processing followed by high-speed mechanical mixing and high vacuum.	2.0		68.98 ± 2.35 (17.3%)	3.17 ± 0.15 (14%)	3.60 ± 0.23 (12.5%)
Park et al., [120]	CNFs	CNFs were dispersed in the epoxy solution through sonication, and residual solvent was removed by vacuum drying. The composites were pre-cured and then post-cured.	2.0 (vol. %)		~22.7 (50%)	~0.85 (77%)	-
Sancaktar et al., [94]	ECNFs	Non-woven ECNFs fabrics were impregnated with epoxy resin.	2.06		~180.0 (20%)	~7.0 (22%)	~3.5 (−12%)
Rana et al., [121]	CNFs	Using a combination of ultrasonication and surfactant.	0.1		55.4 (24.5%)	0.5405 (98.3%)	11.0 (−6%)
Zhang et al., [122]	CNFs	Dispersion of the CNFs/epoxy mixture using a three-roll calender, followed by stirring to blend with the curing agent, and finally undergoing post-curing.	0.125 (vol. %)		~86.0 (6%)	~2.95 (9%)	~5.9 (2%)
Wang et al., [123]	VGCNFs	CNFs mixed with an epoxy resin	1.0	Pristine	~69.0 (−%)	~3.7 (15%)	-
		CNFs were functionalized and dispersed in epoxy using acetone and sonication. The acetone was then removed from the mixture under vacuum.	3.0	Functionalized	~63.0 (22%)	~2.9 (11%)	-
Zhu et al., [108]	CNFs	The CNFs are allowed to wet completely without disturbance, then mechanically stirred, and finally sonicated, all at RT.	0.5	Pure	80.7 (11.6%)	2.9 (11.5%)	5.0 (16.3%)
		Functionalization of CNFs (mechanical stirring at RT) via silanization (ultrasonically stirred) and vacuum drying. The mixing process with the resin is the same as with pure CNFs.	0.3	Functionalized	81.3 (12.4%)	2.1 (−19.2%)	6.8 (58.2%)
Sun et al., [112]	CNFs	CNFs were immersed in dimethylacetamide and CNFs/epoxy nanocomposites were sonicated and mechanically stirred. Evaporation, vacuum degassing followed by curing and post-curing.	1.0		74.4 ± 2.4 (8.3%)	1.22 ± 0.01 (17.3%)	-
Rana et al., [109]	VGCNFs	Combination of ultrasonication and high-speed mechanical stirring.	0.5		~63.0 (65%)	~3.2 (35%)	-
Nie et al., [124]	CNFs	CNFs were dispersed in acetone using an ultrasonic probe, added to the resin and sonicated. The acetone solvent was removed by rotatory evaporation. The CNFs/epoxy composite was placed in a vacuum desiccator followed by curing and post-curing.	1.0	Original	~47.5 (3%)	-	-
		CNFs functionalized by a multistage process including oxidation, reduction, and silanization. The mixing process with the resin is the same as with original CNFs.	0.5	Functionalized	~48.0 (4.1%)	-	-
Chaos-Morán et al., [125]	CNFs	Functionalization of CNFs by oxidation. Untreated or carboxylated CNFs were mixed with epoxy by magnetic stirred, followed by high shear mixing, sonication, the whole process at 40 °C. Finally placed in a vacuum oven for evaporation.	0.5	Original	~70.5 (3%)	~2.55 (1.2%)	-
				Functionalized	~61.0 (−11%)	~2.45 (−2.8%)	-
Rana et al., [115]	VGCNFs	Functionalized CNFs subjected to a bath sonicator followed by magnetic stirring. CNFs/epoxy dispersion using ultrasonication followed by high-speed mechanical stirring.	0.1	Original	61.4 ± 3.2 (59.5%)	2.92 ± 0.02 (24.3%)	-
				Functionalized	62.4 ± 2.6 (62.1%)	2.64 ± 0.06 (12.3%)	-
Shokrieh et al., [126]	CNFs	CNFs/epoxy was mixed at a high shear rate, followed by sonication and degassing the solution in a vacuum chamber.	1.0		-	~3.4 (10.0%)	-
Sánchez et al., [127]	CNFs	CNFs/epoxy dispersion using a three-roll calender, repeating the process several times.	5.0		-	~6.1 (26.0%)	-
Yang et al., [128]	CNFs	Mix CNFs/epoxy several times at temperature followed by ultrasonication. Cure at RT and post-cure in oven.	4.0		52.69 (−14.3%)	2.98 (4.9%)	2.23 (−168.6%)
Shokrieh et al., [97]	VGCNFs	High speed mechanical mixing and sonicated via probe sonicator.	0.25		~74.0 (23%)	~2.75 (10%)	~6.5 (−50%)
Colloca et al., [129]	CNFs	Mixing in a high shear mechanical mixer at temperature followed by degassing. Cure at RT and post-cure in oven.	0.6 (vol. %)		~25.5 (−%)	~2.78 (−%)	-
Chen et al., [90]	ECNFs	Surfaces oxidation and functionalization. The nano-epoxy mixture was first subjected to ultrasonication, followed by mechanical stirring, and degassing and finally post-curing.	0.5		54.39 (22.5%)	-	-
	VGCNFs				52.7 (18.7%)	-	-
	GCNFs				56.6 (27.5%)	-	-
Ahmadi et al., [98]	CNFs	Dispersed in acetone/epoxy resin under mechanical stirring by high-speed, sonicated, and vacuum oven.	1.0		91.87 ± 2.02 (14%)	2.62 ± 0.10 (4%)	4.6 ± 0.43 (−11.5%)
Liu et al., [116]	CNFs	CNFs and deionized water were mixed by magnetic stirring, dopamine was added and the suspension was stirred at RT. CNFs were treated by vacuum filtration and washing. Next, pure CNFs/epoxy and modified CNFs/epoxy composites were mixed by high-speed shear followed by ultrasonic treatment and degassed at RT. Cured at RT and post-cured in oven.	0.4	Pristine	64.7 ± 1.4 (4.7%)	2.608 ± 0.047 (1.9%)	-
				Modified	69.5 ± 1.4 (12.5%)	2.715 ± 0.035 (6.1%)	-
Aziz et al., [130]	CNFs	Pure and modified (amine-functionalized) CNFs were dispersed in acetone by sonication. The epoxy was added and sonicated before the acetone was removed in an oven. The mixture was then placed in a vacuum desiccator and cured at temperature.	5.0	Pristine	~51.0 (6%)	~4.0 (8.1%)	-
			2.0	Modified	~59.0 (22.7%)	~6.2 (67.6%)	-
Wang et al., [117]	CNFs	CNFs/epoxy was mixed with acetone solution using a high-speed mechanical stirrer and sonicated. Then were ultrasonically dispersed, mechanically stirred on temperature, and vacuumed.	0.5		67.0 (22.5%)	2.9 (7.4%)	3.8 (18.2%)
Chanda et al., [131]	CNFs	CNFs/epoxy were mixed by hand, sonicated and degassed in a vacuum oven. It was then placed between parallel aluminum electrodes and an electric field applied to produce a CNFs/epoxy composite aligned in the thickness direction.	0.6	Random	49.9 ± 0.2 (15.8%)	2.539 ± 35.0 (16.5%)	-
			0.4	Aligned	45.3 ± 0.3 (5.1%)	2.418 ± 16.0 (10.9%)	-
Le et al., [111]	CNFs	CNFs and epoxy hardener were hand mixed, followed mixed using ultrasonication. Resin was added to this solution by hand mixing and degassed in a vacuum chamber at RT. Cure at RT and post-cure in oven.	0.1		70.0 (6.4%)	1.1 (10%)	0.14 (25%)

**Table 6 polymers-16-02152-t006:** Summary of studies related to the effect of CNFs on the tensile properties of composite laminates.

Autor, Year, Ref.	Fiber/Matrix	CNFs Type	CNFs Integration Method	Manufacture Process	Optimum Loading (CNFs wt.%)		Tensile Strength [MPa] (Increase (%))	Tensile Modulus [GPa] (Increase (%))	Tensile Strain [%] (Increase (%))
Iwahori et al., [81]	CF/EP	CSNFs	Mechanical mixing, passing through a vacuum chamber and post-cure in a hot press.	Hand lay-up, vacuum application, and post-cure in a hot press.	10.0		577.2 (0.2%)	56.24 (−1.7%)	4.0 (13%)
Bortz et al., [100]	CF/EP	CNFs	Hand mixing and TRM.	VARTM	1.0		165.0 (8%)	11.8 (5%)	11.9 (11.2%)
Rana et al., [109]	CF/EP	VGCNFs	Combination of ultrasonication and high-speed mechanical stirring.	Cured under the heat and pressure in a compression molding machine.	0.5		~790.0 (18%)	~67.0 (37%)	-
Palmeri et al., [132]	CF/EP	CNFs	Shear mixing.	Laid up by hand.	0.67		514.0 (6%)	52.0 (10%)	1.07 (5%)
Sarim et al., [102]	CF/EP	CNFs	CNFs dispersed using a high shear mix, sonicated in a bath ultrasonicator, followed by a spray-up process.	VARTM	1.0		196.0 (22.5%)	12.5 (14%)	-
Rana et al., [115]	CF/EP	VGCNFs	Functionalized CNFs subjected to a bath sonicator followed by magnetic stirring. CNFs/epoxy dispersion using ultrasonication followed by high-speed mechanical stirring.	Hand lay-up and compression molding.	0.1	Original	737.0 ± 69.5 (34.8%)	65.1 ± 4.4 (10.7%)	-
						Functionalized	905.5 ± 11.3 (65.0%)	79.7 ± 2.9 (36.0%)	
Shokrieh et al., [126]	CF/EP	CNFs	CNFs/epoxy was mixed at a high shear rate, followed by sonication and degassing the solution in a vacuum chamber.	Hand lay-up.	1.0		-	~8.3 (12.0%)	-
Zhou et al., [40]	CF/EP	CNFs	High-intensity ultrasonic processor and high-speed mechanical.	VARIM	2.0		617.0 (11%)	46.1 (2.0%)	1.51 (4.1%)
Ahmadi et al., [98]	UHMWPE/EP	CNFs	Dispersed in acetone/epoxy resin under mechanical stirring by high-speed, sonicated, and vacuum oven.	Fibers were impregnated into this resin bath and winded on the metal frame.	1.0		176.3 ± 2.6 (118.7%)	3.42 ± 0.19 (35.7%)	7.03 ± 0.29 (35.2%)
Anjabin et al., [118]	Basalt/EP	CNFs	Functionalized and mixed using an overhead mechanical stirrer.	Hand lay-up, followed by static pressing.	0.3		328.0 (12.7%)	19.8 (32.9%)	-

**Figure 6 polymers-16-02152-f006:**
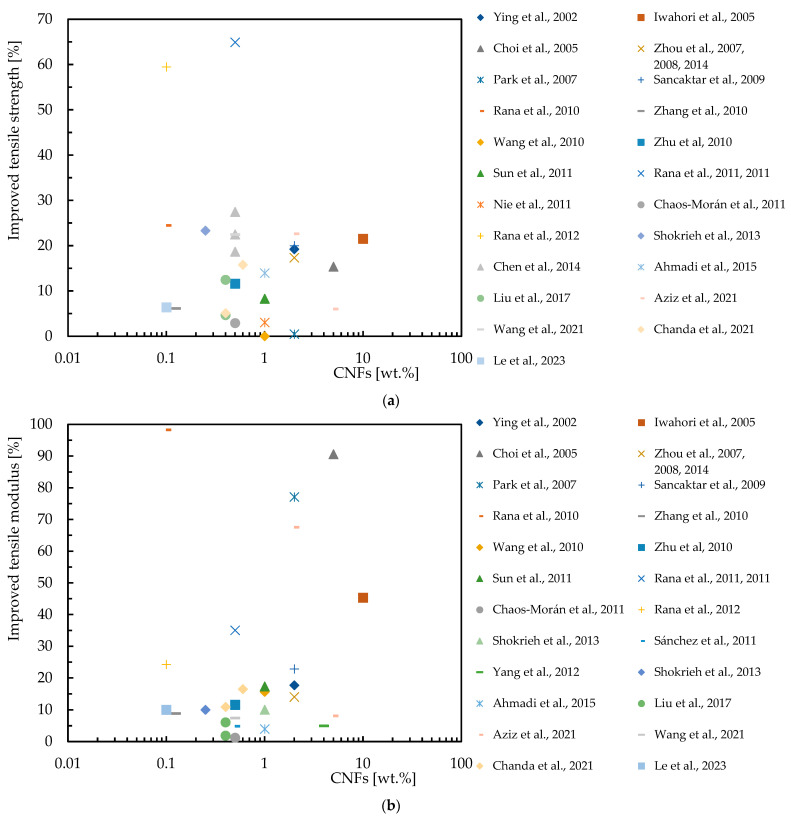
Benefits achieved by epoxy matrix nanocomposites with CNFs in terms of: (**a**) Tensile strength; (**b**) Tensile modulus. Data obtained from [40,81,90,94,97,98,108,109,111,112,113,115,116,117,119,120,121,122,123,124,125,126,127,128,129,130,131].

**Figure 7 polymers-16-02152-f007:**
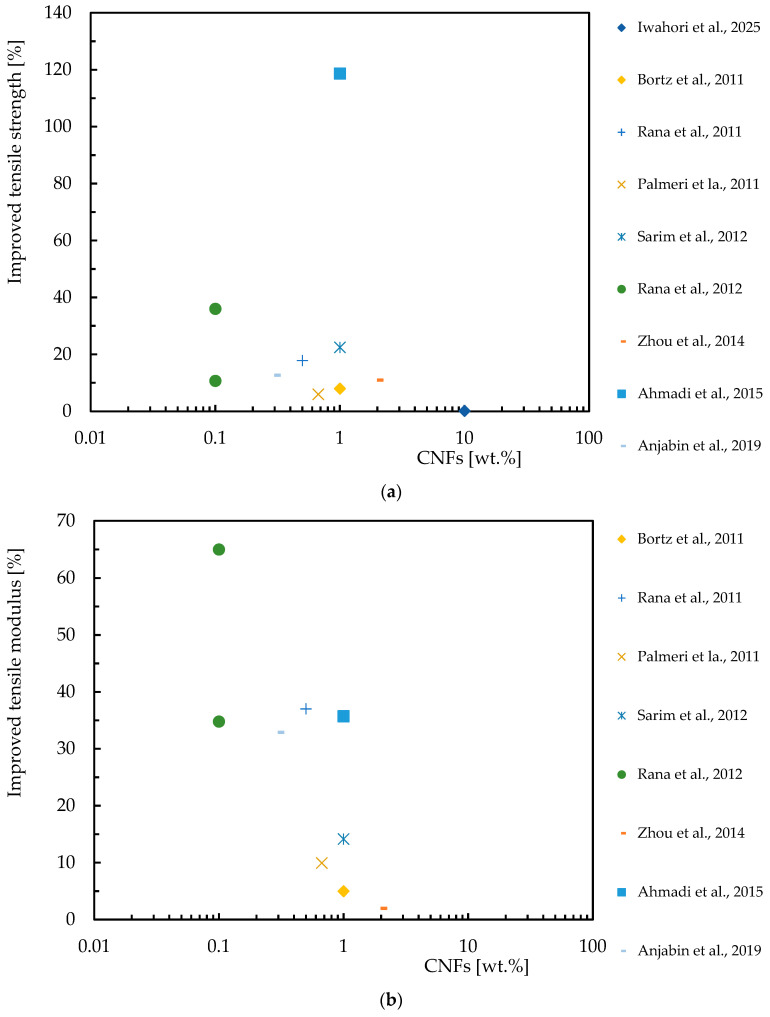
Benefits achieved by composite laminates with CNFs in terms of: (**a**) Tensile strength; (**b**) Tensile modulus. Data obtained from [40,81,98,100,102,109,115,118,126,132].

### 3.2. Effect of CNFs on the Strain Rate Response

Composites can behave differently under dynamic loading compared to static loading. The strain rate response helps to understand how a composite will perform under rapidly applied loads, which is critical in applications such as impact resistance and shock resistance. In industries such as automotive, aerospace, and defense, where composites are often subjected to high strain rates during impacts or explosions, understanding strain rate response is essential to ensure safety and performance. 

The mechanical properties of composites, such as yield strength, tensile strength, and ductility, can vary significantly with strain rate. For example, when composites are subjected to higher strain rates, their stress and modulus increase and their strain decreases. In the design phase, engineers must consider strain rate sensitivity when designing components that will experience different load rates. This will ensure that the selected composites will perform reliably under the expected service conditions.

The speed at which a material deforms is known as its strain rate, and varying strain rates can significantly affect a material’s properties. In this context, the presence of CNFs may contribute to altering the sensitivity of the composite. Although there are not many studies on this topic in the literature, Table 7 summarizes the data in terms of strain rate effect on composites reinforced with CNFs, while Figure 8, Figure 9 and Figure 10 quantify the benefits obtained with this nano-reinforcement in terms of strength and modulus.

Zhou et al. [40,114], for example, studied the tensile response of a nano-enhanced epoxy resin with different contents of CNFs (1.0, 2.0, and 3.0 wt.%) and for strain rates between 0.02 min^−1^ and 2 min^−1^, observing that these nanocomposites are sensitive to the strain rate. For the range of strain rates studied, the authors observed a variation in tensile strength and modulus of around 13.0% and 20.3% for the neat resin and 21.6% and 20.5% for the nanocomposite reinforced with 2.0 wt.% of CNFs, respectively. Poveda et al. [133] studied the compressive response of syntactic foams reinforced with CNFs for strain rates between 1.7 × 10^−6^ and 50 min^−1^, and observed that the strength and modulus increased by around 7.3% and 15.5%, respectively, compared to the values obtained for the control conditions. According to Shokrieh et al. [134] the influence of adding 0.25 wt.% of VGCNFs on the tensile mechanical properties of the epoxy matrix at dynamic strain rates of 0.000028 min^−1^ leads to increases in tensile modulus and strength of 12.40% and 11.03%, respectively. Zhou et al. [135] studied the dynamic response to compression of epoxy composites functionalized with 0.75 wt.% CNFs for different loading rates (0.00005 to 60 min^−1^) and obtained, for example, for a strain rate of 60 min^−1^, an increase in Young’s modulus and compressive strength of around 57% and 195.5%, respectively, compared to the neat epoxy matrix. Kar et al. [107] studied the influence of loading rates on the bending performance for a temperature of 30 °C, and observed that adding 1.0 wt.% of CNFs to the laminate led to an increase in bending strength of around 13% and 14% and in bending modulus of around 9% and 18%, respectively, for 1 and 100 mm/min, due to the improvement promoted by CNFs in terms of stress transfer and a reduction in the rate of crack propagation. Finally, Santos et al. [136], studied two resins with different viscosities nano-reinforced with CNFs for strain rates between 0.0014 and 1.8 min^−1^, achieving improvements of between 4% and 12% compared to the bending strength obtained for the neat matrix and of between 3% and 12% in terms of the bending modulus. Similar results were obtained for carbon laminates nano-reinforced with CNFs and it was concluded that the maximum bending stress and bending modulus increase with increasing strain rate for all laminates [36].

**Table 7 polymers-16-02152-t007:** Summary of studies related to the effect of CNFs on the strain rate response of epoxy matrix nanocomposites.

Autor, Ref.	Fiber/Matrix	CNFs Type	CNFs Integration Method	Manufacture Process	Optimum Loading (CNFs wt.%)	Strain Rate
Zhou et al., [114]	-/EP	CNFs	High-intensity ultrasonic processing followed by high-speed mechanical mixing and high vacuum.	-	2.0	Tensile strength increase, tensile modulus increase, and failure strain decrease to neat and with 2.0 wt.% CNFs/EP to 0.02 min^−1^, 0.2 min^−1^, and 2 min^−1^ velocities.
Poveda et al., [133]	-/EP	CNFs	A mechanical mixer fitted with a high shear impeller was used, the mixture placed on a shaker for degassing and curing at RT.	-	1.0 to 10.0	The compressive strength and modulus under quasi-static testing increase.
Shokrieh et al., [134]	-/EP	VGCNFs	High speed mixing, sonication, and final degassing of the mixture.	-	0.25	Tensile strength and tensile modulus increase to neat EP and tensile strength decrease and tensile modulus increase to 0.25 wt.% CNFs/EP to 0.00167 min^−1^, 0.1 min^−1^, and 0.2 min^−1^ velocities.
Chanda et al., [131]	-/EP	CNFs	CNFs/epoxy were mixed by hand, sonicated and degassed in a vacuum oven. It was then placed between parallel aluminum electrodes and an electric field applied to produce a CNFs/epoxy composite aligned in the thickness direction.	-	0.4	The elastic modulus and tensile strength increased with increasing strain rates, to aligned composites and random composites, however, transversely aligned composites, compared to random composites, always exhibited lower modulus, strength and failure strain, under strain rates of 0.001, 0.01, 0.085, and 0.17 min^−1^.
Zhou et al., [135]	-/EP	CNFs	Functionalization, mixture subjected to magnetic stirring and vacuum in an oven. Cured and post-cured at high temperatures.	-	0.75	To 0.00005 to 60 min^−1^ Young’s modulus and compressive strength increases.
Kar et al., [107]	GF/EP	CNFs	Magnetic stirring dispersion, ultrasonication, and degassing.	Hand lay-up, followed hot pressing.	1.0	CNFs composite increases bending strength and bending modulus of GF with CNFs composite at 30 °C when tested at 1 and 100 mm/min loading rate
Santos et al., [136]	EP	CNFs	Simultaneous dispersion in a high-speed shear mixer at high shear rate and sonication for 3 h at RT followed by degassing.	-	0.75	Independently of the resin and weight percentages of the CNFs, both materials are strain-rate sensitive when subjected to bending strain rates of 0.00015 to 1.15 min^−1^. The bending stress and modulus increase for higher values of strain rate values.
					0.5	
Santos et al., [36]	CF/EP	CNFs	Simultaneous dispersion in a high-speed shear mixer at high shear rate and sonication for 3 h at RT followed by degassing.	Hand lay-up and simultaneous application of vacuum and pressure.	0.75	Laminates produced with CNFs nano-reinforced resins show a greater sensitivity to strain rate than the corresponding control laminates by approximately 10% for bending strength and 3% to 13% for bending modulus.
					0.5	

**Figure 8 polymers-16-02152-f008:**
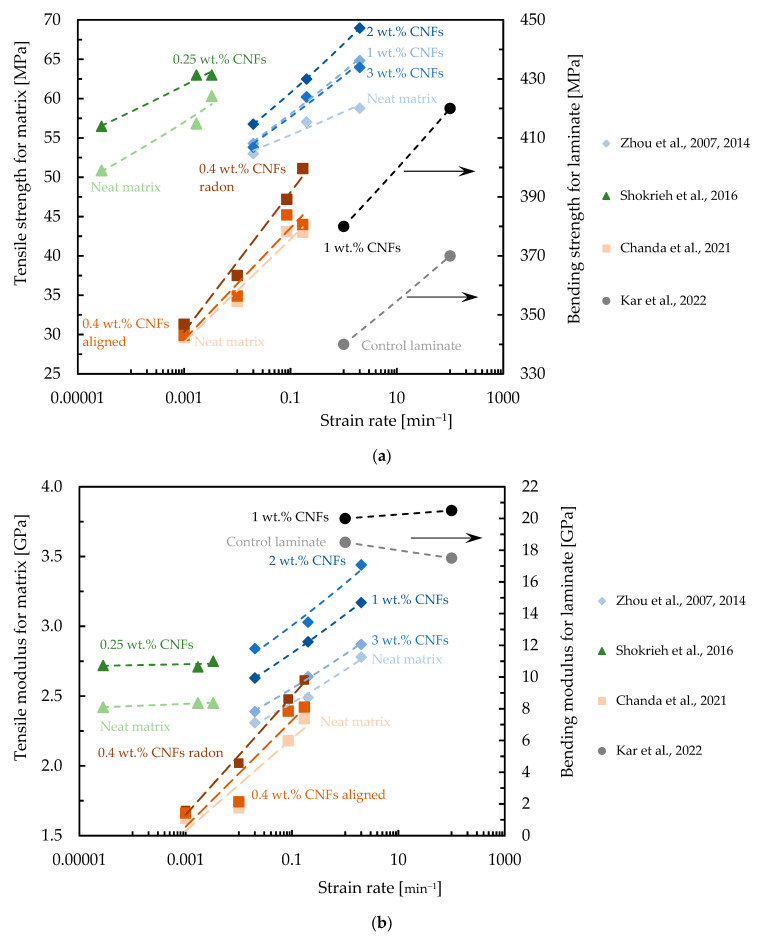
Benefits achieved with CNFs in terms of: (**a**) Tensile strength and bending strength; (**b**) Tensile modulus and bending modulus. Data obtained from [40,107,114,131,134].

**Figure 9 polymers-16-02152-f009:**
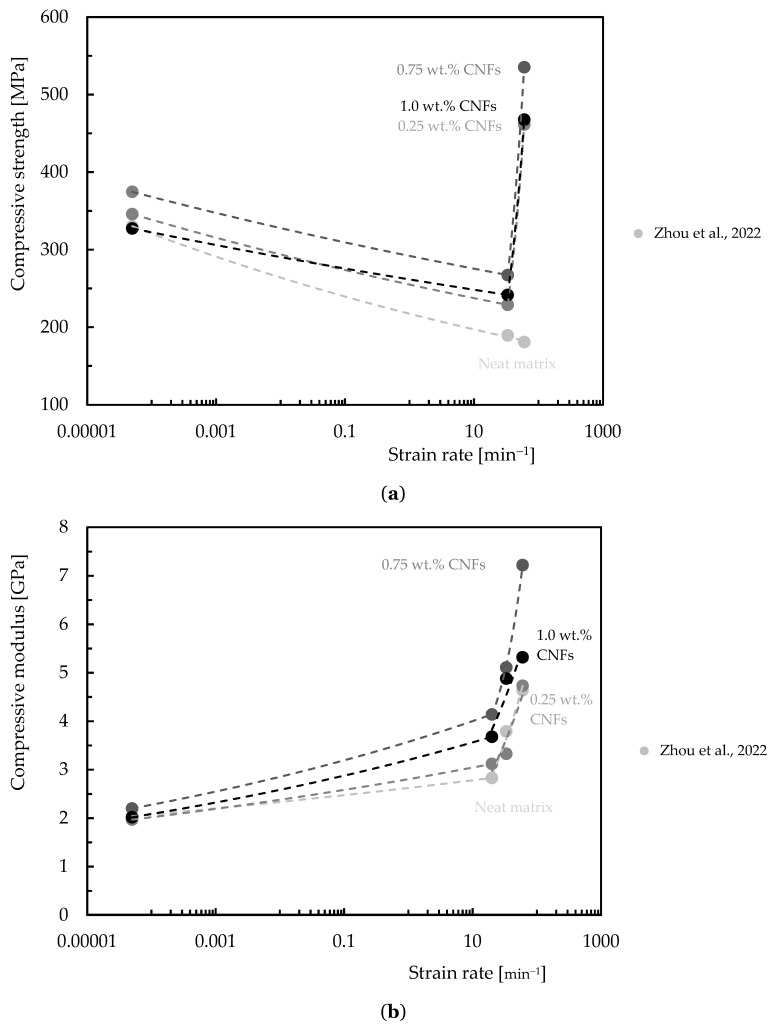
Benefits achieved with CNFs in terms of: (**a**) Compressive strength; (**b**) Compressive modulus. Data obtained from [135].

**Figure 10 polymers-16-02152-f010:**
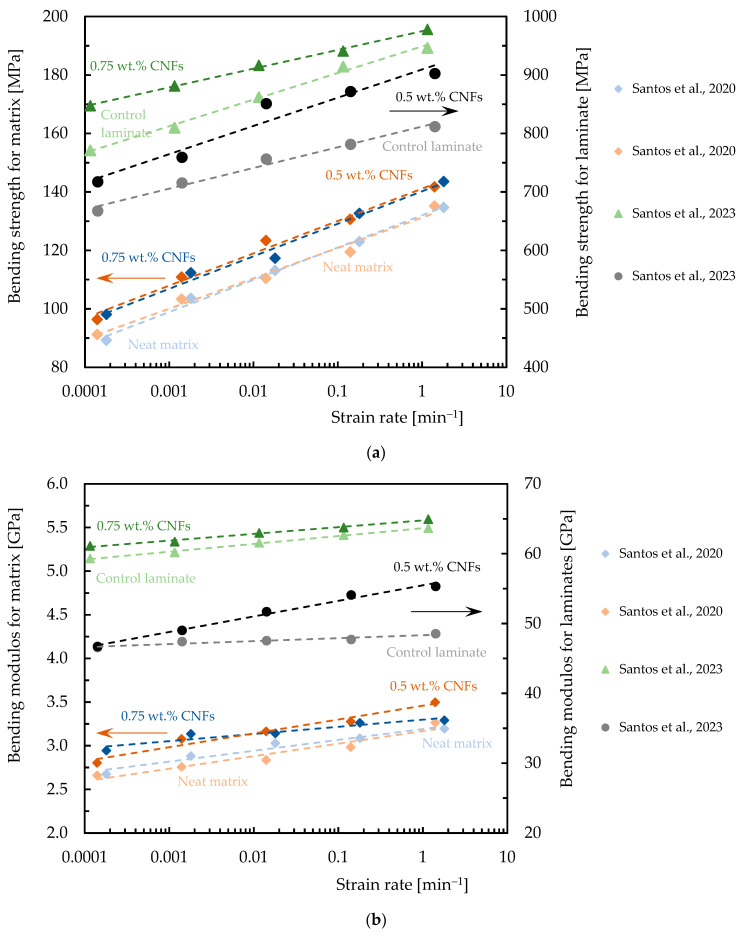
Benefits achieved with CNFs in terms of: (**a**) Bending strength; (**b**) Bending modulus. Data obtained from [36,136].

### 3.3. Effect of CNFs on Stress Relaxation and Creep Behavior

In engineering applications, understanding the stress relaxation and creep phenomena is very important, especially when it is necessary to guarantee the long-term dimensional stability of structures/components produced by polymer-based composite. However, most published studies focus essentially on characterizing the mechanical behavior of these complex systems in terms of static properties. However, similar to the previous topic, this subject was also not extensively studied for epoxy-based composites reinforced with CNFs; nevertheless, the available studies are summarized in Table 8, while Figure 11 and Figure 12 quantify the benefits obtained.

**Table 8 polymers-16-02152-t008:** Summary of studies related to the effect of CNFs on the stress relaxation and creep response of epoxy matrix nanocomposites and composite laminates.

Autor, Ref.	Fiber/Matrix	CNFs Type	CNFs Integration Method	Manufacture Process	Optimum Loading (CNFs wt.%)	Stress Relation	Creep
Santos et al. [34]	EP	CNFs	Simultaneous dispersion in a high-speed shear mixer at high shear rate and sonication for 3 h at RT followed by degassing.	-	0.75	Stress relaxation and creep behavior were shown to be strongly dependent on the applied load level and the addition of CNFs significantly reduces both phenomena.	
					0.5		
Santos et al. [137]	CF/EP	CNFs	Simultaneous dispersion in a high-speed shear mixer at high shear rate and sonication for 3 h at RT followed by degassing.	Hand lay-up and simultaneous application of vacuum and pressure.	0.75	The addition of CNFs may not have a good effect on the stress relaxation and creep behavior of the laminates.	
					0.5		

Santos et al. [34,137] studied the effect of CNFs on the stress relaxation and creep behavior of two epoxy resins with different viscosities. Regardless of the resin, the authors observed that the viscoelastic response is strongly dependent on the load levels applied, but this influence decreases with the introduction of CNFs. In terms of stress relaxation, for example, for a constant displacement corresponding to bending stress of 80 MPa, the nano-reinforcement of the matrix with CNFs promoted decreases of around 21.7% and 9.2% compared to the neat resin with lower viscosity and higher viscosity, respectively, while for the creep response, the decreases observed were around 12.8% and 47.6%, respectively. Considering the laminates produced with these resins, the displacements corresponding to the highest applied loads decreased by around 6.6% and increased ~21%, respectively, compared to the laminates produced with neat resin with lower viscosity and higher viscosity, while the creep displacement decreased by about 11.8% and increased ~12.8%, respectively. In all cases, for the lowest loading levels, no advantage was discernible for resins filled with CNFs.

When CNFs are added to the epoxy matrix, the stress relaxation behavior of the nanocomposites decreases due to the additional resistance of molecular rearrangement that restricts the movement of the polymer chain. Nevertheless, the influence of CNFs on stress relaxation is more pronounced at higher load levels due to the improved load transfer capability.

CNFs increase the stiffness of the epoxy matrix, which helps reduce the overall creep strain. The increased stiffness provided by the CNFs limits the deformation of the composite under sustained loads. The reinforcement limits the ability of the polymer chains to move, thereby reducing the time-dependent deformation. Experimental data show that nanocomposites with higher CNF content exhibit significantly lower creep strain compared to neat epoxy matrix. The incorporation of CNFs improves the dimensional stability of the composites under long-term loading conditions, making them more suitable for structural applications.

### 3.4. Effect of CNFs on the Interlaminar Shear Strength (ILSS)

Composites with higher fiber-matrix adhesion, higher strength, and higher matrix toughness are desired because they can be subjected to high stresses during their service life, which can lead to crack propagation through the fiber-matrix interfaces. In order to evaluate this mechanical performance, it is usual to carry out short beam shear (SBS) tests to assess the interlaminar shear strength (ILSS) of the composite. In this case, the shear force resulting from sliding between layers of the composite or its deformation between them is obtained.

Failure may not occur at the mid-plane, because it is difficult to ensure pure shear during [138]; however, failure results from a combination of different failure modes, such as microcracks in the resin, indentation, fiber breakage, micro-buckling, bending, and interlaminar shear cracking of the specimens [139,140]. It should be noted that the competition between failure mechanisms depends on the quality of the polymer matrix, the morphology of the fiber surface (smooth or rough), and the bonding mechanism between the fiber and the matrix [106]. However, many studies available in the literature report benefits when CNFs are incorporated into the matrix, which are summarized in Table 9, while their quantification is shown in Figure 13.

Green et al. [88], for example, found increases of 23% when they nano-reinforced the matrix with 0.1 wt.% of CNFs in an E-glass/epoxy composite, but for 1.0 wt.%, the ILSS values obtained worsened due to the agglomerates formed or poor dispersion resulting from the manufacturing method used. Similarly, Bortz et al. [100] also observed a decrease compared to the control laminates, in this case of around 4%, when 1.0 wt.% of CNFs were added. Chen et al. [101] found an increase in ILSS by incorporating ECNFs mats of around 86.2% compared to the control laminate, which represented an increase from 27.5 MPa to 51.2 MPa. Palmeri et al. [132] studied unidirectional and quasi-isotropic composites and observed that, in both cases, the ILSS increased when CNFs were added. In the case of unidirectional composites, the improvement was around 15% with the addition of 0.69 wt.% of CNFs, while in quasi-isotropic composites it was 22% with the addition of 0.67 wt.% of CNFs. The studies carried out by Rodriguez et al. [141] showed that the incorporation of CNFs modified by deposition on the surface of the carbon fiber led to an improvement in the ILSS compared to composites reinforced with untreated fibers. In this case, the composite containing oxidized CNFs exhibited a 9.08% increase in ILSS, while the panels reinforced with amidized CNFs, deposited on both sized and unsized fibers using multiscale reinforcement fabrics (MRFs), showed even greater improvements. The ILSS increased by 10.01% for the panel with amidized CNFs deposited on sized fibers and by 12.44% for the panel with amidized CNFs deposited on unsized fibers. The addition of oxidized CNFs and amidized CNFs in these panels was 0.67 wt.% and 1.0 wt.%, respectively. Sarim et al. [102], obtained an increase of around 25% by incorporating 1.0 wt.% CNFs into laminates manufactured using the VARTM process.

In studies carried out by Khan et al. [142,143], the authors first impregnated the CNF sheets separately with the polymer and then integrated them into the composite. This method aimed to eliminate the weak wetting of nanofillers when present together with fibers/fabrics on a microscale in RTM or VARTM processes. By impregnating the CNF sheets with polymer beforehand, the polymer could be infused throughout the bulk nanofiller–FRPCs composite, ensuring better compatibility and distribution of the nanofillers within the composite structure. Experimental studies developed by Chen et al. [103] showed that the deposition of electrospun precursor nanofibers on carbon fiber fabric leads to an improvement in ILSS of around 221.1% compared to the control laminates (i.e., an increase from 27.5 MPa to 88.3 MPa). Annad et al. [144] found that adding 0.5 wt.% CNFs to epoxy composites led to increases of 33% in ILSS, using the resin film infusion (RFI) technique. The studies performed by Chen et al. [90] showed that the inclusion of small contents of CNFs (between 0.1 wt.% and 0.3 wt.%) in the epoxy resin promotes significant improvements in the ILSS. Zhou et al. [40] used the VARIM technique to produce unidirectional composites with 2.0 wt.% CNFs and obtained an increase of around 15.8% in the ILSS compared to the control composites. Lake et al. [44] produced carbon/epoxy laminates, and the highest increase in ILSS, about 14.5%, was observed for the composite incorporating 5.0 wt.% of CNFs.

Ma et al. [145] used the filter membrane-assisted method to produce carbon/epoxy composites reinforced with CNFs and observed the highest increase in ILSS of around 55% for 3.0 wt.% of CNFs, while in the non-membrane-assisted method, the maximum increase in ILSS achieved was 11% and obtained for 1.0 wt.% of CNFs. Dhakate et al. [35], reinforced only the interlaminar region between the fabric layers with 1.1 wt.% of CNFs, which led to an increase in ILSS of around 190% compared to the control composite. Yao et al. [146] used 0.4 wt.%-modified CNFs and obtained a 73% increase in ILSS. Ranabhat et al. [147] showed that the ILSS can increase by around 24.2% when a CF/epoxy composite is reinforced with 0.5 wt.% CNFs. Finally, Santos et al. [36] produced carbon/epoxy laminates with different types of epoxy resin, varying their viscosity, and observed that the improvement in ILSS values varied between 8.6% and 9.4%. In this study, they also observed that the ILSS was sensitive to the strain rate (i.e., higher strain rate higher ILSS values) and the presence of CNFs reduces the corrosive effect on the interlaminar shear strength of the composites. 

Failure of composites is often caused by destruction of the interface. Improving the bond strength is critical to the preparation of high-performance composites. Incorporating CNFs into the epoxy matrix increases its strength and improves the interface, thereby increasing the stress transfer and consequently the ILSS of the composites. When CNFs are sufficiently and effectively oriented along the thickness direction and stitched into the interfiber space, they can modify the distribution of interlaminar shear stress and increase the ILSS. This improvement in interlaminar properties is driven by the roughness induced by the nanofillers, which increases the wettability of the fibers and results in excellent mechanical interlocking between the fiber and the matrix. The improvement in ILSS with the addition of CNFs can be attributed to the nanoscale reinforcement provided by the presence of CNFs. These nanofibers bridge microcracks and prevent crack front propagation. The high aspect ratio and large surface area of CNFs facilitate effective stress transfer between the matrix and the fibers. In addition, CNFs can improve the toughness of the matrix, increasing its ability to absorb energy and delay delamination [79,106,148].

**Table 9 polymers-16-02152-t009:** Summary of studies related to the effect of CNFs on the ILSS of composite laminates.

Autor, Ref.	Fiber/Resin	CNFs Type	CNFs Integration Method	Manufacture Process	Optimum Loading (CNFs wt.%)	ILSS [MPa] (Increase (%))
Quaresimin et al., [149]	CF PrP/EP	VGCNFs	Dispersed in the EP system according to an attrition milling process.	Hand lay-up of prepreg. Curing performed using a vacuum bag and additional pressure.	7.5	~50.0 (~14%)
Green et al., [88]	E-GF/EP	CNFs	Mechanical mixer.	VARIM and compress.	0.1	43.85 ± 1.0 (23.6%)
Bortz et al., [100]	CF/EP	CNFs	Hand mixing and TRM.	VARTM	1.0	~50.0 (−4%)
Chen et al., [101]	CF/EP	ECNFs	Fabrication of mats of ECNFs.	VARTM with interlaminar regions containing mats of ECNFs.	~2.5	51.2 ± 4.9 (86.2%)
Palmeri et al., [132]	CF/EP	CNFs	Shear mixing.	hand placed	0.67	132.3.0 (15%)
Rodrigues et al., [141]	CF/EP	CNFs	Oxidized CNFs (O-CNFs) and amidized CNFs (A-CNFs).	VARTM	1.0	59.58 (12.4%)
Khan et al., [142]	CF/EP	CNFs	Simple soaking, hot compression molding, and vacuum infiltration.	Bucky paper interleaves.	10.0	~70.0 (31%)
Miranda et al., [89]	CF/EP	CNFs	CNFs grown onto the surface of CF fabrics.	Hot pressed	0.2	~33.0 (−10%)
Arai et al., [150]	CF PrP/EP	VGCNFs	CNFs were inserted between prepregs layers using a sifter.	Autoclave	Area density of 10 [g/m^2^]	52.2 (24.9%)
Sarim et al., [102]	CF/EP	CNFs	CNFs dispersed using a high shear mix, sonicated in a bath ultrasonicator followed by a spray-up process.	VARTM	1.0	~375.0 (25%)
Khan et al., [143]	CF/EP	CNFs	Simple soaking, hot compression molding, and vacuum infiltration.	Bucky paper interleaves.	10.0	~69.0 (31%)
Chen et al., [103]	CF/EP	ECNFs	Thermal treatments of stabilization in air followed by carbonization in argon.	VARTM	14.0	88.3 ± 5.8 (221.1%)
Anand et al., [144]	E-GF/EP	CNFs	Mechanical probe sonicator and mechanical mixing.	RFI	0.5	83.6 ± 0.52 (33.1%)
Chen et al., [90]	CF/EP	ECNFs	Surfaces oxidation and functionalization. The nano-epoxy mixture was first subjected to ultrasonication, followed by mechanical stirring, and degassing and finally post-curing.	VARTM	0.3	45.8 ± 7.1 (42.2%)
		VGCNFs				38.3 ± 3.5 (18.9%)
		GCNFs				37.4 ± 1.3 (16.1%)
Zhou et al., [40]	CF/EP	CNFs	High-intensity ultrasonic processor and high-speed mechanical.	VARIM	2.0	41.6 ± 1.7 (15.9%)
Lake et al., [44]	CF/EP	CNFs	Producing a nanofiber mat composed of highly graphitic CNFs in an isotropic array.	VARTM	5.0	2250 (14.5%)
Ma et al., [145]	CF/EP	CNFs	High intensity ultrasonic atomizer probe and mechanical mixing.	Filter membrane-assisted.	3.0	64.0 (50.9%)
				No filter membrane-assisted.	1.0	45.5 (7.3%)
Srikanth et al., [151]	CF/EP	CNFs	Probe ultrasonicator followed by ball milling and aminofunctionalized.	Fabric layers were impregnated and compressed.	1.0	41.0 ± 1.1 (28.1%)
Taheri-Behrooz et al., [152]	E-GF/EP	CNFs	Mixed and stirred, then sonicated using probe sonicator.	Vacuum-assisted hand lay-up.	0.25	44.76 ± 0.28 (19.5%)
Dhakate et al., [35]	CF PrP/EP	CNFs	Mixed and sonicated.	Impregnated and was applied temperature and pressure (hot plate).	1.1	55.0 (103.7%)
Kirmse et al., [153]	CF PrP/EP	CNFs	Flow-transferring a resin film containing electrical-field aligned CNFs.	Autoclave-vacuum bag.	1.0	53.93 (35.1%)
Yao et al., [146]	CF/EP	VGCNFs	Synthesis and spraying of the polymergrafted VGCNFs functionalized.	Degassed under vacuum, hot compressed.	0.4	83.0 ± 8 (72.9%)
Anjabin et al., [118]	Basalt/EP	CNFs	Functionalized and mixed using an overhead mechanical stirrer.	Hand lay-up, followed by static pressing.	0.3	80.2 (73.6%)
Kirmse et al., [154]	CF PrP/EP	CNFs z-threads	Shear mixing	Autoclave	0.85	44.81 (50.1%)
De et al., [106]	CF/EP	CNFs	Electrophoretic deposition (EDP) technique.	Hand lay-up, followed hot pressing.	0.5 [g/L]	~36.0 (16%)
Kirmse et al., [155]	CF PrP/EP	CNFs	High-sheared, sonicated, and degassed mixture.	Non-isothermal flow-transfer process.	1.0	69.72 ± 2.51 (7.4%)
Ranabhat et al., [147]	CF/EP	CNFs	Radial flow alignment technique.	Out-of-autoclave vacuum-bag-only (with 20% acetone in resin to create voids).	0.5	54.5 (24.2%)
Ravindran et al., [41]	CF/EP	CNFs	Hand-mixing followed by a TRM.	High pressure compression molding process.	1.0	39.0 ± 1.9 (−5.6%)
He et al., [148]	CF/EP	CNFs	TRM	Multilayer resin film infusion-compressive molding (with 10 min infusion).	0.3	53.65 (5.4%)
Yao et al., [79]	CF/EP	CNFs	Chemical vapor deposition.	Vacuum and temperature.	-	72.1 (18.6%)
Mrzljak et al., [156]	CF/EP	CNFs	Mechanical stirring under a vacuum and milled in a TRM and E-field was application to align the CNFs.	Hand-layup process and pressed in a hot press applied during curing.	0.7 random	55.5 ± 3.5 (6.1%)
Santos et al., [36]	CF/EP	CNFs	Simultaneous dispersion in a high-speed shear mixer at high shear rate and sonication for 3 h at RT followed by degassing.	Hand lay-up and simultaneous application of vacuum and pressure.	0.75	55.4 ± 1.7 (8.6%)
					0.5	54.0 ± 3.2 (9.3%)

**Figure 13 polymers-16-02152-f013:**
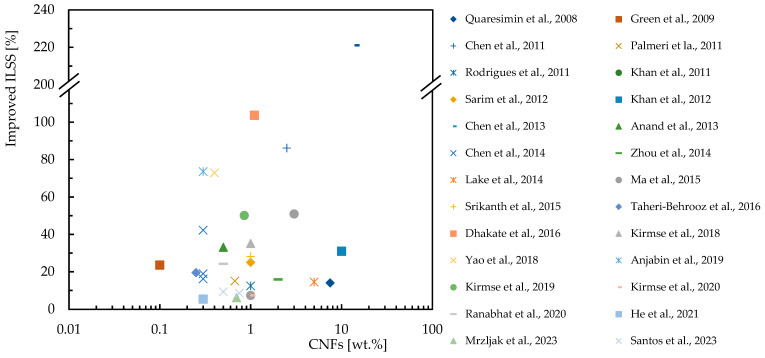
Benefits achieved with CNFs in terms of ILSS. Data obtained from [35,36,40,44,88,90,101,102,103,118,132,141,142,143,144,145,146,147,148,149,151,152,153,154,155,156].

### 3.5. Effect of CNFs on the Mode I and Mode II Interlaminar Fracture

Fracture toughness, or the resistance of the composite interface to crack propagation, is an important material property that is generally associated with three different failure modes: mode I, which is characterized by out-of-plane tensile crack initiation; mode II, characterized by in-plane shear loading; and mode III, describes out-of-plane shear loading. For typical service conditions, pure mode loading is extremely rare and generally occurs in mixed mode I/II. Therefore, to overcome the weakness of these materials in many industrial applications, there are several studies in the literature that report the benefits achieved through the use of CNFs, which are summarized in Table 10, and the improvements obtained are quantified in Figure 14.

According to Arai et al. [157], for example, incorporating 20 g/m^2^ of CNFs into the interlayer of a unidirectional carbon/epoxy composite can increase the mode I fracture toughness by around 50% compared to control laminates, while the mode II fracture toughness was 2 to 3 times higher. Zhu et al. [158] carried out a similar study incorporating 0.5 wt.% of modified CNFs in specific areas of the resin layer, especially in the region surrounding the fracture plane and along the delamination growth path, and obtained improvements of 57% and 49% for mode I and mode II, respectively. In the study performed by Yokozeki et al. [159], authors obtained improvements in interlaminar fracture toughness using modified CNFs. Gude et al. [160] used 0.25 wt.% CNFs and found improvements of around 10% in *G_IC_*. Kostopoulos et al. [161] obtained a 100% increase in fracture energy by incorporating 1.0 wt.% CNFs into the composite matrix due to the extensive fiber bridging promoted by the CNFs. Ladani et al. [162] and Ravindran et al. [38,163] observed that incorporating CNFs with through-thickness fiber rod reinforcements, such as z-pins, can promote synergistic effects on interlaminar fracture toughness. In this case, the combination of multiple reinforcements can promote additional toughening mechanisms that are not present in the case of single reinforcements. In another study, Ladini et al. [164] also concluded that CNFs are effective in increasing delamination resistance with just 1.0 wt.% of CNFs.

**Table 10 polymers-16-02152-t010:** Summary of studies related to the effect of CNFs on the *G_IC_* and *G_IIC_*.

Autor, Ref.	Fiber/Matrix	CNFs Type	CNFs Integration Method	Manufacture Process	Optimum Loading (CNFs wt.%)		*G_IC_* [kJ/m^2^] (Increase (%))	*G_IIC_* [kJ/m^2^] (Increase (%))
Kostopoulos et al., [165]	CF/EP	CNFs	-	Hand lay-up followed autoclave process.	1.0		~0.8 (100% by MBT); ~1.0 (100% by areas method)	~2.6 (50% by MBT); ~2.2 (57% by areas method)
Kostopoulos et al., [166]	CF/EP	CNFs	Mixing and vacuum.	Hand lay-up followed autoclave process.	1.0		0.91 (133.3% by MBT); 1.0 (100% by area method);	-
Tsantzalis et al., [167]	CF/EP	CNFs	Temperature and vacuum.	Hand lay-up followed autoclave process.	1.0		~0.8 (100% by MBT); ~1.0 (100% by areas method)	-
Quaresimin et al., [149]	CF PrP/EP	VGCNFs	Dispersed in the EP system according to an attrition milling process.	Prepeg hand lay-up. Curing using a vacuum bag between a platen press under vacuum and additional pressure.	7.5		~0.09 (initiation decrease ~ 55%); ~0.14 (propagation decrease ~ 70%)	~1.5 (over 100%)
Arai et al., [157]	VGCF PrP/EP	VGCNFs	VGCNFs/EP interlayer: VGCNFs/ethanol mixed manually and dispersed using a roller.	Autoclave	20 [g/m^2^]		~0.65 (23.8%)	~0.28 (100%)
Li et al., [75]	VGCF PrP/EP	VGCNFs	Powder method (applied in the middle plane by hand lay-up process).	Autoclave	20 [g/m^2^]		0.432 (95.5% critical load); 0.616 (26% fracture resistance)	-
Yokozeki et al., [159]	CF/EP	CSCNFs	CSCNFs dispersed EP (sprinkle) and CSCNFs dispersed film between layers (planetary mixer, and dispersed using the wet mill with ceramic beads).	Hand lay-up followed autoclave.	5.0 wt.% CSCNFs-dispersed EP with 10.0 wt.% CSCNFs-dispersed film between layers.		0.227 (167%)	1.753 (208.6%)
Bortz et al., [100]	CF/EP	CNFs	Hand mixing and TRM.	VARTM	1.0		~0.42 (35%)	-
Gude et al., [160]	CF PrP/EP	CNFs	Dispersion	Autoclave	0.5		0.0991 ± 0.0077 by area method; 0.096 ± 0.0087 by CBT; 0.0967 ± 0.0082 by ECM; (~15% for all methods)	
Kostopoulos et al., [161]	CF/EP	VGCNFs	Mixing and declassification by applying vacuum.	Hand lay-up and cured in an autoclave, using the vacuum bag technique.	1.0		0.79 (100% by MTB); 1.002 (100% by areas method)	2.626 (86.4% by MTB); 2.195 (55.8% by areas method)
Palmeri et al., [132]	CF/EP	CNFs	Shear mixing	Hand placed	0.67		~1.40 (decrease ~ 20%)	-
Khan et al., [142]	CF/EP	CNFs	Simple soaking, hot compression molding and vacuum infiltration.	Bucky paper interleaves.	10.0		-	~2.49 (104%)
Hu et al., [76]	CF PrP/EP	VFCNFs	Manually dispersed (zigzag pattern) using the powder method.	Autoclave	20 [g/m^2^]		0.432 (95.5% critical load); ~0.62 (~30% fracture resistance)	-
Zhu et al., [158]	S2-GF/EP	CNFs	Functionalized: magnetically stirred, sonicated in an ultrasonic bath with temperature.	Wet filament winding method and hot pressing.	0.5	Neat	0.165 ± 0.014 (onset 30%); 0.903 ± 0.0015 (propagation 47%)	0.783 ± 0.0037 (onset 39%); 0.996 ± 0.0067 (propagation 46%)
						Functionalized	0.176 ± 0.0084 (onset 39%); 0.968 ± 0.0041 (propagation 57%)	0.843 ± 0.011 (onset 49%); 0.963 ± 0.0023 (propagation 41%)
Koissin et al., [168]	CF/EP	CNFs	Infusion	Hand lay-up	2.6		~1.1 (crack start ~ 95%); ~0.9 (crack stop 140%)	-
Arai et al., [169]	CF/EP	VGCNFs	planetary centrifugal mixer.	VARTM	10 [g/m^2^]		~0 55 (20%)	-
Wang et al., [170]	CF/EP	CNFs	CNFs functionalized, sonication, and vacuum applied at the end.	Vacuum and hot pressure applied.	0.5		~0.30 (13.6%)	~0.51 (21.7%)
					1.0		~0.29 (9%)	~0.61 (45.3%)
Ma et al., [145]	CF/EP	CNFs	High intensity ultrasonic atomizer probe and mechanical mixing.	Filter membrane-assisted.	3.0		-	~0.815 (~90%)
Ladani et al., [162]	CF/EP	CNFs	TRM to disperse and E-field application.	Cured at RT.	1.6		2.345 (1650%)	-
Wu et al., [171]	CF/EP	VGCNFs	Magnetic stirring to functionalise the CNFs (ultraprobe sonication and simultaneous stirring). Sonication of the EP and subjected to a magnetic field.	Joints bonded using the Fe_3_O_4_ at CNFs/EP were cured at RT.	0.4		0.328 (144.8% aligned)	-
							0.242 (80.6% random)	
Gude et al., [172]	CF PrP/EP	CNFs	EP adhesive was dispersed in chloroform and mixed by ultrasonication.	Two surface treatments applied to the laminates: grit blasting and atmospheric plasma and cured in the autoclave with vacuum bag.	0.5		~0.090 (~10% peel-ply)	-
							0.2755 ± 0.0091 (26.5% grit blasted)	
							0.1739 ± 0.0361 (4.7% plasma)	
Hsiao et al., [173]	CF PrP/EP	CNFs	Magnetic stirrer, high shear mixing followed by agitation in a sonicator and then degassing.	Hand wet lay-up process and placed on a hot plate.	0.3		0.348 (14%)	-
Ladani et al., [174]	CF/EP	CNFs	TRM	Wet hand-layup process and cured at RT in the hydraulic press.	1.0		1.123 (67.6%)	-
Ladani et al., [175]	CF/EP	CNFs	TRM and E-field application.	DCB joints as a 2 mm thick adhesive layer bonding.	2.0		2.16 (1490% random)	-
							2.27 (1570% aligned)	
Wu et al., [176]	E-GF/EP	VGCNFs	TRM, hand lay-up process and E-field application.	Vacuum bag, the matrix was then cured at RT.	0.7	Initiation toughness:	~0.6 (50% random) ~0.8 (100% aligned)	-
						Steady state toughness:	~1.5 (25% random); ~2.16 (80% aligned)	
Ladani et al., [164]	CF PrP/EP	CNFs	TRM and E-field application.	Autoclave	1.0		1.29 ± 0.112 (862.7% random)	-
							1.642 ± 0.161 (1125% aligned)	
Ravindran et al., [177]	CF PrP + E-GF PrP/EP	CNFs	TRM and E-field application.	Autoclave	1.0		1.260 (830% random)	-
Ladani et al., [178]	CF PrP/EP	CNFs	TRM dispersion.	Nano EP adhesive layer was cured at RT.	2.0		~2.16 (1570%)	-
Ravindran et al., [163]	CF/EP	CNFs	Hand mixing and TRM.	Liquid compression molding.	1.0		1.40 ± 0.12 (91%)	2.88 ± 0.24 (42%)
Ravindran et al., [38]	CF/EP	CNFs	CNFs hand-mixed into the EP and passed four times through a TRM.	Wet compression molding process (consolidation in a hydraulic press).	5.0		0.84 ± 0.15 (240% initiation toughness)	-
							2.04 (179.5% steady state toughness)	
Ravindran et al., [179]	CF/EP	CNFs	Hand-mixed into liquid EP, then passed four times through a TRM.	Liquid compression molding approach.	5.0		-	3.39 ± 0.14 (66%)
Ekhtiyari et al., [180]	E-GF/EP	CNFs	High speed mechanical stirring and ultrasonic agitation.	Hand lay-up process.	0.25		~0.70 (13.5%)	-
Ravindran et al., [39]	CF PrP/EP	CNFs	CNFs hand-mixed into the ER and passed four times through a TRM.	Wet-hand lay-up process and vacuum.	1.0		1.40 ± 0.12 (91.8%)	2.88 ± 0.24 (41.2%)
Kavosi et al., [181]	CF PrP/EP	CNFs	Functionalized	VARTM	2.0		~0.5 (25%)	-
Ravindran et al., [41]	CF/EP	CNFs	Hand-mixing followed by a TRM.	High pressure compression molding process.	1.0		1.40 ± 0.12 (91% initiation toughness)	2.88 ± 0.24 (42% initiation toughness)
							0.55 (~120% steady state toughness)	1.25 (~5% steady state toughness)

**Figure 14 polymers-16-02152-f014:**
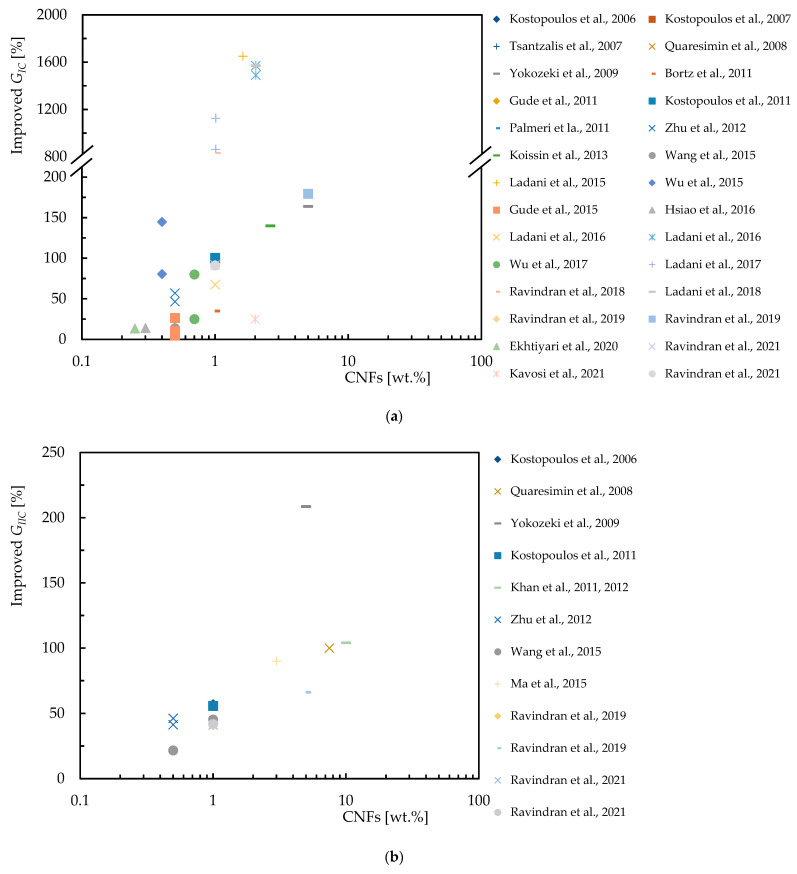
Benefits achieved with CNFs in terms of: (**a**) *G_IC_*; (**b**) *G_IIC_*. Data obtained from [38,39,41,100,132,142,143,145,149,158,159,160,161,162,163,164,165,166,167,168,170,171,173,174,175,176,177,178,180,181].

### 3.6. Effect of CNFs on the Low-Velocity Impact Response

In service, composite structures can be subjected to out-of-plane impact loads, either from natural factors (e.g., falling tree branches, hail, and bird strikes) or from other factors (e.g., falling tools during maintenance operations or impact from small foreign bodies), which can result in the separation of adjacent layers, known as delamination [182]. The composites’ behavior under impact loading, particularly low-velocity impact, is a very complex phenomenon because different damage mechanisms compete simultaneously, such as fiber breakage, delaminations, and matrix cracking, among others [183]. However, nano-reinforced composites with CNFs have several advantages, as shown in Table 11, and quantified in Figure 15 in terms of absorbed energy and respective damage area.

**Figure 15 polymers-16-02152-f015:**
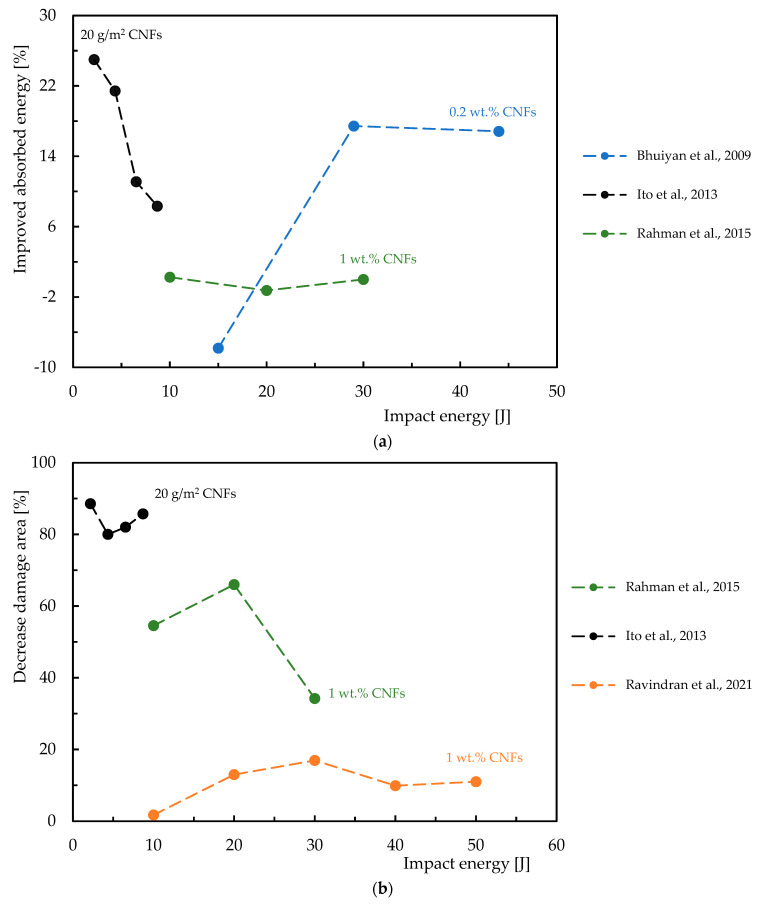
Benefits achieved with CNFs in terms of: (**a**) Absorbed energy; (**b**) Damage area. Data obtained from [41,184,185,186].

**Table 11 polymers-16-02152-t011:** Summary of studies related to the effect of CNFs on the low-velocity response.

Autor, Ref.	Fiber/Matrix	CNFs Type	CNFs Integration Method	Manufacture Process	Optimum Loading (CNFs wt.%)	Impact Energy [J]	Energy to Peak Load Increase [%]	Energy to Max Load Increase [%]	Absorbed Energy Increase [%]	Damage Area Characteristics
Bhuiyan et al., [186]	Polyurethane sandwich CF/EP face sheets	CNFs	Sonication, high-speed mechanical mixer.	VARTM	0.2	44	7.7	46.8	16.8	For 0.2 wt.% CNFs, 20.3% less base width damaged and 18.0% less indentation depth, compared to the control sandwich.
Ito et al., [185]	CF PrP/EP	VGCNFs	VGCNFs interlayer.	Autoclave	20 [g/m^2^]	8.69	-	-	−8.3	The damaged area is much smaller when VGCNFs interlayer is inserted between the carbon laminate layers.
Arai et al., [187]	CF/EP	CNFs	Planetary centrifugal mixing.	Hand lay-up, vacuum, and hot cure.	2.5	8.69	-	-	−5.6	Less delamination area for the lamination with 2.5 wt.% CNFs
Rahman et al., [184]	CF PrP/EP	CNFs	CNFs oxidized, mechanical stirrer, sonicated, and degasification at RT.	OOA-VBO	1.0	30	~19.0	-	0	The damage area decreased with the incorporation of O-CNFs at all impact energy levels and a maximum reduction of 67% in damage area is obtained at 20 J.
Ivañez et al., [188]	Sandwich structures with woven CF/EP face-sheets and Nomex	CNFs	Mechanical stirrer and simultaneously ultrasonic bath.	Manual filling of the damage.	0.75	10 to 30	In repaired sandwich structures with CNFs/EP is higher than in baseline specimens.	In repaired sandwich structures is less than in baseline specimens.		To the impact energy of 10 J no damage was identified on repaired sandwiches and at an impact energy of 30 J, some cracks appear around the area of impact.
Ravindran et al., [41]	CF/EP	CNFs	Hand-mixing followed by a TRM.	High pressure compression molding process.	1.0	10 to 50	~4.0 increase to 50 J impact energy	-	-	For impact energy of 10, 20, 30, 40 and 50 J there was a reduction in the damaged area in the order of 1.5%, 13%, 17% 9.9% and 11%, in the nano laminates reinforced with 1.0 wt.% CNFs in comparison with the control laminate.

Bhuiyan et al. [186], for example, observed that reinforcing foam cores with 0.2 wt.% CNFs led to an improvement in fracture toughness and a reduction in the damaged area. Ito et al. [185] introduced an intermediate layer of 20 g/m^2^ of CNFs between unidirectional prepregs and observed that the damaged area was much smaller than that of the control laminates. Rahman et al. [184] analyzed the response of composites reinforced with CNFs, and for a content of 1.0 wt.% the absorbed energy decreased and, consequently, the damaged area. For example, reductions of damaged area around 30%, 70%, and 58% were obtained for impacts with energies of 10 J, 20 J, and 30 J, respectively.

## 4. Discussion

The analysis shows that CNFs are excellent reinforcements for an epoxy matrix, as long as the dispersion methods and processing techniques are appropriate. In fact, it is essential to ensure: (i) the uniformity of the dispersion of the CNFs in the matrix and, subsequently, their distribution among the reinforcing fibers; (ii) avoidance of the presence of agglomerates of CNFs, which result in the concentration of tensions and the formation of voids in the composites; (iii) the degree of CNF breakage; (iv) the mixing of the CNFs directly into the resin, which can lead to high viscosities that create problems during processing; and (v) the interface between the CNFs/matrix/reinforcement, which is crucial for the efficiency of load transfer and overall mechanical performance [189]. In addition, the magnitude of the improvements achieved varies substantially depending on the intrinsic characteristics of the epoxy resin used (physical, chemical, and mechanical properties), and the CNFs (length/diameter ratio, geometry, and mechanical and chemical properties due to surface alterations).

The incorporation of CNFs into an epoxy matrix can enhance its mechanical properties due to several mechanisms, those being (i) interfacial interaction: Strong interfacial adhesion between CNFs and the epoxy matrix is crucial for effective load transfer. Surface treatments or functionalization of CNFs can enhance this adhesion, further improving mechanical properties [190]; (ii) aspect ratio of CNFs: The length-to-diameter ratio of CNFs plays a crucial role in reinforcing the epoxy matrix. Higher aspect ratio provides a larger surface area for stress transfer between the matrix and the CNFs, enhancing the composite’s mechanical properties. Long CNFs can bridge cracks more effectively, improving toughness and resistance to fracture [191]; (iii) dispersion: Uniform distribution of CNFs within the epoxy matrix are vital for maximizing mechanical performance. Well-dispersed CNFs ensure consistent reinforcement throughout the matrix, preventing weak points that could lead to premature failure. Poor dispersion promotes agglomerates, which act as stress concentrators and reduce the composite’s strength [110,192,193]; (iv) alignment of CNFs: This can significantly influence the mechanical properties of the composite. Aligned CNFs provide directional reinforcement, improving properties such as tensile strength and stiffness along the alignment direction. Randomly oriented CNFs offer isotropic reinforcement, but typically result in lower mechanical enhancement compared to aligned configurations [171,175]; (v) voids and porosity: Voids between CNFs and the matrix, often due to poor adhesion or processing issues, can detract from the composite’s mechanical properties. These voids create sites for crack initiation and propagation, reducing strength and toughness. Vacuum-assisted processing and careful control of curing parameters can help minimize void content [193]; and (vi) processing techniques: The methods used to process the CNF-reinforced epoxy composites have a significant impact on their final properties. In addition to the techniques associated with the manufacture of composites, in the manufacture of fiber reinforced polymers, the good dispersion is critical to achieving the desired mechanical performance. Indirect processes that use fibers pre-impregnated with matrix (e.g., vacuum bag/autoclave molding and compression molding) can influence the distribution, alignment, and adhesion of CNFs within the matrix. On the other hand, direct processes that use separate fibers and matrix that are brought together at the point of molding (e.g., wet lay-up combined with a vacuum bag, and VARTM) can minimize voids and ensure uniform dispersion and alignment of the CNFs, optimizing the mechanical properties of the composite.

In terms of static response, and in particular, bending, the focus is on CNF content of 0.25 wt.% to 1.0 wt.% and whose improvements provided increases in bending strength of between 10% and 416.8% and bending modulus of between 0.5% and 143.6%. Although fewer in number, there are also studies that report benefits for higher CNF contents [80,81,82], between 3 wt.% and 19.2 wt.%, but involving resins with certain specificities (such as viscosity). In this case, the improvements obtained were between 16.8% and 95.3% and 27.7% and 97.2% in terms of bending strength and bending modulus, respectively. 

The research results show that increasing the CNF content in epoxy matrices leads to a significant improvement in mechanical properties such as flexural strength and flexural modulus. This improvement can be explained by the increased number of bonding points and better dispersion of the filler in the composite matrix. As the CNF content increases, it provides a denser and more effective load transfer network, resulting in a more uniform distribution of applied stresses and greater resistance to deformation. 

Scientifically, the correlation between CNF content and mechanical properties can be attributed to the interaction between the CNFs and the epoxy matrix. The CNFs act as reinforcements that prevent movement of the polymer chains, thereby increasing the stiffness and strength of the material. In addition, the interface between the CNFs and the epoxy matrix creates strong bond zones that can withstand high loads without premature failure.

Concerning the composite laminates, it is important to reinforce the matrix because its mechanical properties are much lower than those of the fibers used. In this context, most studies refer to CNF content of between 0.3 wt.% and 1.0 wt.%, for which average improvements in bending strength and bending modulus are 20% and 10%, respectively. Regarding the tensile properties of the nanocomposites, the main reinforcements are in the range of 0.4 wt.% and 1.0 wt.%, where the gains in tensile strength and modulus of elasticity were between 3.0% and 64.9% and 1.2 and 98.3%, respectively. In terms of laminates, a range of CNF contents between 0.3 wt.% and 1.0 wt.% is observed, and improvements of between 6.0% and 108.7% in tensile strength and between 5.0% and 37% in modulus of elasticity are achieved.

Polymer-based composites are sensitive to the strain rate, for which the stress and modulus increase with increasing the strain rate, while the strain decreases [194,195]. However, when the CNFs are introduced into the matrix, it is possible to observe that the composite becomes less sensitive to the strain rate. This reduction can be attributed to several factors, those being (i) improved load transfer: CNFs improve the load transfer between the polymer matrix and the reinforcing fibers. This means that under varying strain rates, the stress distribution within the composite becomes more uniform, reducing the overall sensitivity to strain rate changes; (ii) increased stiffness and strength: The addition of CNFs enhances the stiffness and strength of the composite. Higher stiffness and strength generally mean that the material is less likely to exhibit large variations in mechanical properties with changing strain rates; (iii) enhanced energy absorption: CNFs can increase the energy absorption capacity of the composite. This leads to better resistance to deformation under rapid loading conditions, thereby reducing the strain rate sensitivity; and (iv) microstructural changes: The presence of CNFs can alter the microstructure of the resin, improving its ability to resist deformation. This includes modifications at the molecular level that enhance the overall stability of the composite under dynamic loading.

In the context of viscoelastic response, namely stress relaxation, physical processes typically involve minimal formation or breakage of primary bonds, resulting in molecular rearrangements. On the other hand, due to chemical processes, it involves chain scission, crosslink scission, or crosslink formation [196]. Thus, CNFs are added, and these establish a network that contributes to restricting the mobility of the polymer chains [197]. Nonetheless, these benefits depend on efficient load transfer combined with good dispersion of the nanoparticles [108]. In the case of load transfer, it essentially depends on interfacial interactions, including weak van der Waals interactions between the filler materials and the matrix polymer, potential chemical bonds through treatments, and mechanical interlocking due to irregular fiber surfaces [198,199]. Furthermore, when polymers are reinforced with long fibers, these obstruct molecular flow, leading to a delay in the relaxation process [200]. Additionally, interface properties play a crucial role because relaxation processes may arise from bond breakage and subsequent propagation. In composites, stress relaxation is a result of two mechanisms: relaxation within the matrix phase and the occurrence of fiber-matrix debonding zones followed by crack propagation [201]. 

In terms of creep, for example, this phenomenon in polymers occurs even at room temperature and at low-stress levels due to the molecular motion within the backbone polymer arrangement [202]. For neat matrices, creep results from a combination of viscous flow and elastic deformation [203]. According to Bouafif et al. [204], molecular motions within the backbone polymer arrangement lead to the creep phenomenon that is influenced by stress levels. Jian et al. [205] even propose a quantitative relationship between molecular mobility and macroscopic deformation. Therefore, the addition of a relatively low content of nanoparticles restricts the mobility of the epoxy matrix polymer chains and, consequently, prevents the chains’ disentanglement and slippage [205]. In this case, creep phenomena are slowed down (decrease the creep displacements), although varying load concentrations can cause contradictory effects [34,137]. CNFs contribute to immobility by impeding slippage, realignment, and movement of the polymer chains [197]. This effect is attributed to three main mechanisms: (i) the establishment of robust interfacial strength between the CNFs and the matrix; (ii) the CNFs functioning as obstruction sites; and (iii) the high aspect ratio of the CNFs [201,206]. However, similar to stress relaxation, increasing the weight content of CNFs can affect its effectiveness in the creep response of a composite due to agglomeration problems and deterioration in filler/epoxy adhesion [205]. Moreover, in composites reinforced with long fibers, the creep process is delayed [207], but strongly influenced by the fiber orientation [208].

Therefore, it is possible to conclude that although there is not an abundance of studies on the effect of CNFs on the stress relaxation and creep response of composites with epoxy-based matrices, the literature provides numerous studies involving other types of nanoparticles, such as graphene, CNTs, or nanoclays, that can mitigate the knowledge about the viscoelastic phenomenon of these materials [209,210,211].

To promote good fiber/matrix adhesion, there are three ways to improve it [79]: (i) matrix toughening could improve the in-plane and interlaminar toughness of composites simultaneously, but it also introduces changes in the viscosity, *Tg*, and thermal properties of the resin, which would affect the manufacturing process of CFRP composites; (ii) z-direction toughening, such as Z-pin and stitching, forms bridging structures in the interlaminar region of the composites to achieve an apparent toughening effect, but the in-plane properties would be reduced to some extent; and (iii) the in-plane performance of composites using the 3D weaving method is significantly lower than that of typical laminates, but the long experimental process cycle, complex operating procedures, and relatively high manufacturing costs limit their use in practical applications.

The advantage of using CNFs is their ability to interconnect and stress transfer at the fiber-matrix interface [138]. From this review most of the results obtained focus on the addition of CNFs between 0.3 wt.% and 1.0 wt.%, which leads to improvements in ILSS between 5.4% and 103.7%. In the same context, and with regard to delamination fracture resistance, various strategies are mentioned, such as [179]: (i) the incorporation into the matrix of high hardness particles (such as elastomeric or thermoplastic fillers) or carbon, used alone or together (hybrid nano-reinforcements); (ii) the use of different fibers or different orientations, as well as the modification of the fibers’ surface; (iii) the insertion of interleaves or veils composed of thermoplastic, nanofibrous, aramid, or other high-toughness materials between the layers; and (iv) macro-scale through the thickness fiber reinforcement methods, such as stitching, pinning, and orthogonal weaving, are employed to facilitate crack bridging within composite materials. Alternatively, the literature reports the use of nano-particles, such as CNFs with a higher aspect ratio, placed in intermediate layers to the fiber-reinforced plies [157,158]. The results show that the use of CNT between 0.5 wt.% and 1.0 wt.% leads to G_IC_ improvements of between 4.7% and 1650%, while for G_IIC,_ the improvements were between 21.7% and 208.6%. The high dispersion of the results depends on the type of constituents (epoxy matrix, reinforcing fibers, and CNFs) and the manufacturing method.

Finally, in terms of the benefits obtained for low-velocity impact strength, all those mentioned above explain that the introduction of nano-reinforcements is favourable to reducing the damaged area. From the studies available in the literature, it can be seen that the CNFf content generally varies between 0.2 wt.% and 1.0 wt.%, which reports an increase of 8.3% to 25.0% in terms of absorbed energy and 1.7% to 88.6% in terms of damaged area. The CNFs offer several significant benefits when it comes to improving the impact resistance of composites, such as (i) enhanced energy absorption: CNFs can absorb and dissipate impact energy more effectively. This leads to a reduction in the size of the damaged area upon impact; (ii) improved toughness: the addition of CNFs enhances the toughness of the composite material, which is crucial under impact conditions; (iii) reduction in crack propagation: can prevent the growth and propagation of cracks, stopping or slowing down the propagation of cracks, maintaining the overall integrity of the composite under impact; (iv) increased interfacial strength: The interface between the CNFs and the matrix is crucial for load transfer. Improved interfacial bonding due to the presence of CNFs enhances the overall mechanical performance of the composite, particularly under dynamic loading conditions such as impacts; and (v) reduction in delamination: the CNFs help to bridge the layers of the composite, reducing the likelihood of delamination and thereby enhancing the material’s service life. 

## 5. Conclusions

The design of high-performance composites for structural liability applications with higher safety requirements led to the use of nano-reinforcements. In particular, the incorporation of a very small content of CNFs into the epoxy matrix showed significant potential for improving the mechanical properties of the composites. 

In static response, even the addition of 0.25 and 1 wt.% of CNFs can delay crack initiation and reduce the crack propagation, improving the strength and the young modulus of the composite. This mechanical behavior improvement is attributed to the CNFs, which can transfer a significant fraction of the load from the epoxy matrix (the lowest strength element of the composite) to the fibers of the reinforcing fabric. Thus, a homogeneous dispersion of CNFs forms a continuous network within the epoxy matrix and good interfacial adhesion between the CNFs and the epoxy resin can more effectively transfer the loads. In addition, this uniform CNFs network can effectively restrict the movement of the polymer chains. 

The Interlaminar shear strength (ILSS), and fracture toughness are closely related to the fiber-matrix bond strength, the quality of the polymer matrix, and the surface morphology of the fiber (smooth or rough). The incorporation of up to 1 wt.% of CNFs results in improved mechanical response explained by fracture surfaces that reveal extensive fiber bridging (due to the addition of CNFs). Together with fiber-reinforced fabrics, they can even result in Z-shaped “pins”, promoting synergistic effects on the interlaminar fracture properties (both mode I and mode II) of CFRP composites. Furthermore, as a result of the different length scales of the (nano)reinforcements, additional toughening mechanisms are reported.

In addition, the composites showed improved resistance to low-velocity impact (LVI), and the damaged areas were significantly reduced with the incorporation of a small content of CNFs (typically up to 1 wt.%), which is beneficial for dynamic loading conditions. The addition of the CNFs increases the fracture energy, since crack propagation can be limited by the formation of bridges between the epoxy matrix and the CNFs, and therefore better adhesion between them due to the interaction of cross-links. It is likely that the uniformly dispersed CNFs started out as a barrier to crack propagation under lower load conditions. During the crack propagation process, it is slowed down by the pulling force of the nanofibers well impregnated into the crack surface due to an energy dissipation phenomenon and, consequently, the energy of the crack tips is significantly reduced. In addition, there are reports that the crack tips are forced to stop or frequently change their crack line of propagation due to the presence of CNFs. As a result, crack initiation and propagation become difficult in the nano-reinforced matrix, which leads to greater strength and smaller areas of damage in the composites.

Regarding viscoelastic behavior, the inclusion of CNFs reduces time-dependent deformation and improves the structural stability of the composite. Just 1 wt.% CNFs has a detrimental effect on the mobility of the polymer chain in the epoxy matrix, as well as in chain disentanglement and slippage. This way, in terms of stress relaxation and creep resistance, the CNFs can contribute to the reduction in their values. A relatively low amount of well-dispersed CNFs forms interphases that bind to the matrix through bonding segments and junctions, reinforcing the load-bearing capacity and improving the immobility of the polymer chains, preventing sliding and realignment. 

It is clear that the addition of the optimum content of CNFs in a given epoxy matrix, up to 1.0 wt.%, is advantageous, as observed in Figure 4, Figure 5, Figure 6, Figure 7, Figure 8, Figure 9, Figure 10, Figure 11, Figure 12, Figure 13, Figure 14 and Figure 15. Industrial-scale production of composites with CNFs is more feasible than with CNTs or GP, but manufacturing technology, and especially dispersion, is a factor that cannot be neglected.

## Figures and Tables

**Figure 1 polymers-16-02152-f001:**
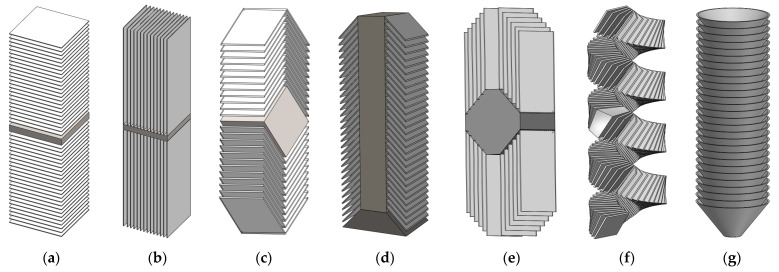
Schematic representation of CNFs and their precipitated graphite platelets formed during the growth for the different morphologies: (**a**) platelet; (**b**) ribbon-type; (**c**) fishbone/herringbone with solid core; (**d**) fishbone/herringbone with hollow core; (**e**) ribbon/herringbone with solid core; (**f**) spiral type; and (**g**) stacked cup. Adapted from [22,47,48,49].

**Figure 2 polymers-16-02152-f002:**
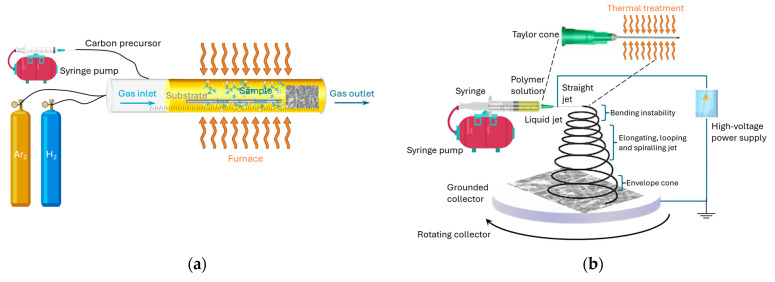
Schematic diagram setup of: (**a**) CCVD [28,53]; (**b**) electrospinning/electrospun [24,28,54]; (**c**) PECVD [64]; (**d**) gas-phase flow catalytic method; (**e**) templating [58,65]; (**f**) phase separation [28]; (**g**) arc discharge deposition [66]; and (**h**) FC [61,67] (schemes adapted from respective references indicated).

**Figure 3 polymers-16-02152-f003:**
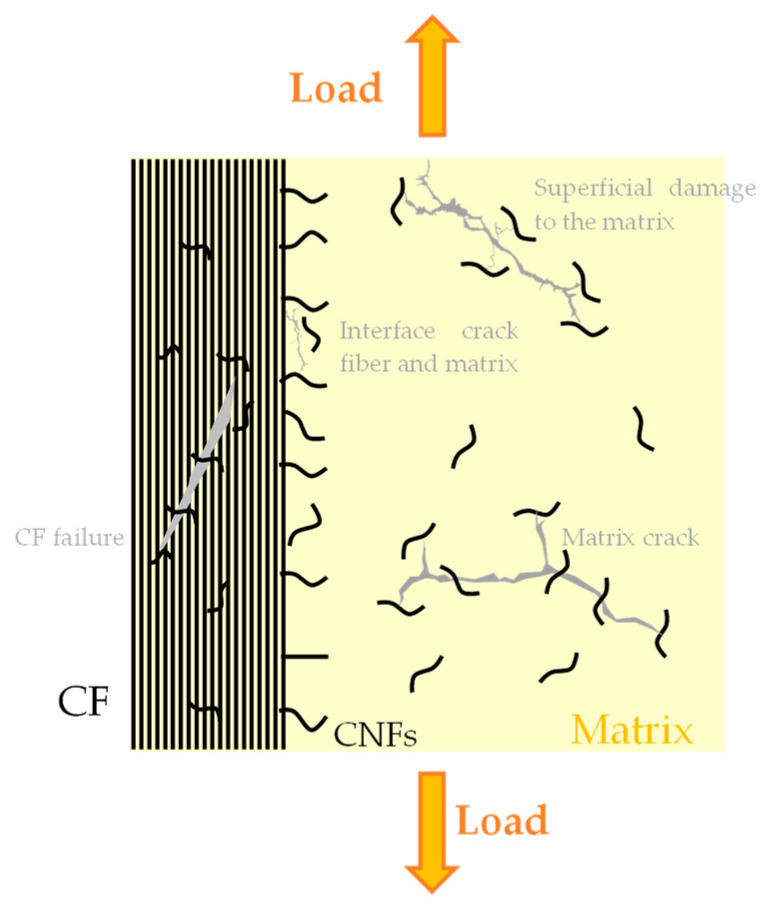
Effects of CNFs on the mechanical properties of epoxy matrix composites. Adapted from [79].

**Figure 11 polymers-16-02152-f011:**
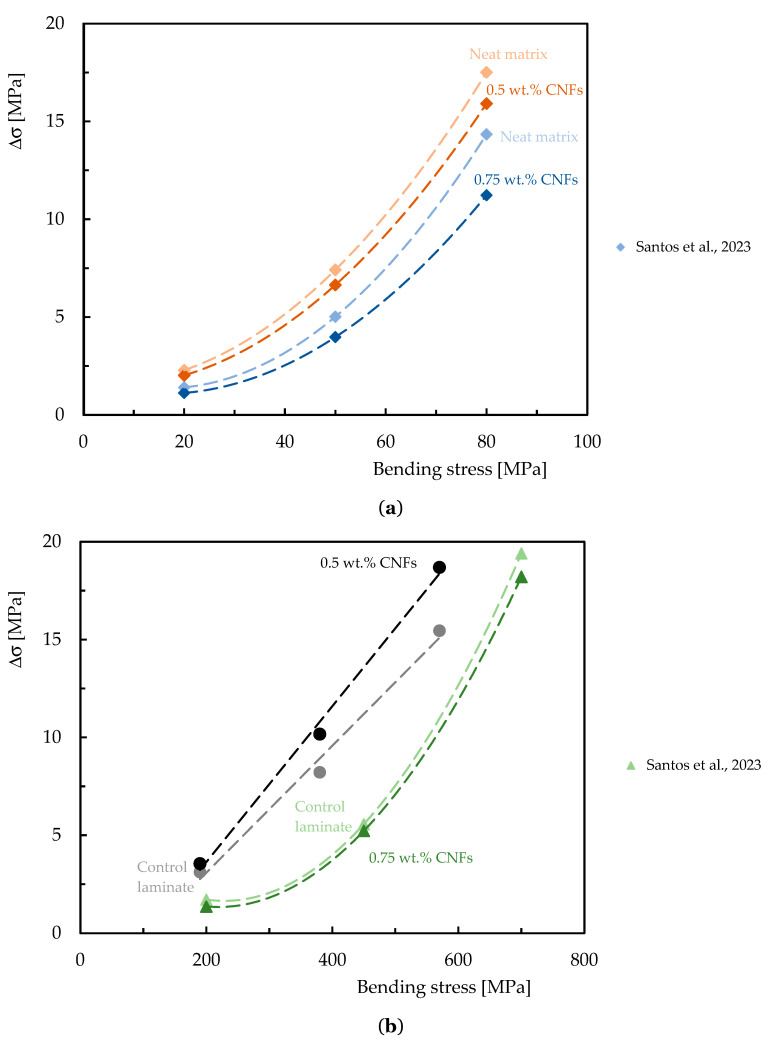
Benefits achieved with CNFs in terms of stress relaxation for: (**a**) Nanocomposites; (**b**) Composite laminates. Data obtained from [34,137].

**Figure 12 polymers-16-02152-f012:**
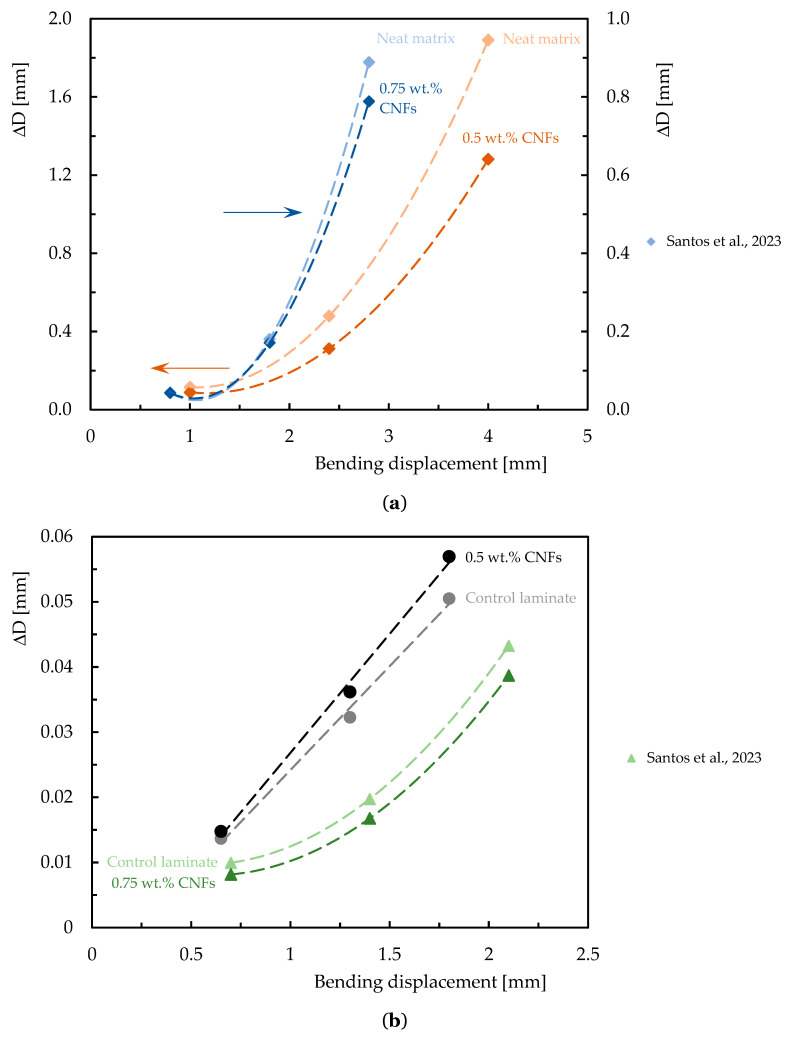
Benefits achieved with CNFs in terms of creep response for: (**a**) Nanocomposites; (**b**) Composite laminates. Data obtained from [34,137].

**Table 1 polymers-16-02152-t001:** Comparison of epoxy matrix with other common thermosetting and thermoplastic matrices on various performance criteria [9,10].

Property	Epoxy Matrix	Other Thermosets Matrix	Thermoplastic Matrix
Mechanical strength	High tensile and compressive strength	Moderate to high (e.g., polyesters: moderate, phenolics: high)	Generally lower than thermosets (varies by type, e.g., nylon: high, polyvinyl chloride (PVC): moderate)
Flexural strength	High	Moderate to high	Varies, often lower than epoxies
Modulus of elasticity	High	Varies (phenolics and vinyl esters: high)	Lower than thermosets
Toughness	High, good fracture toughness	Moderate to high	Varies (e.g., acrylonitrile butadiene styrene (ABS): high, PE: moderate)
Adhesion	Excellent, bonds well to many substrates	Varies (polyesters: moderate, phenolics: high)	Generally lower, requires surface treatment
Chemical resistance	Excellent	Varies (phenolics: high, polyesters: moderate)	Good for specific chemicals (e.g., Polytetrafluoroethylene (PTFE): excellent)
Thermal stability	High, good heat resistance	Moderate to high (phenolics: high, polyesters: moderate)	Lower than thermosets (e.g., polyether ether ketone (PEEK): high, PP: low)
Electrical insulation	Excellent	Moderate to excellent (phenolics: high, polyesters: moderate)	Varies, some are good insulators (e.g., PE: excellent)
Thermal conductivity	Moderate, can be enhanced with fillers	Generally low, can be enhanced with fillers	Varies, some are low (e.g., PE: low)
Shrinkage	Low during curing	Moderate (polyesters: high, phenolics: low)	Low (due to lack of curing)
Dimensional stability	High	Moderate to high	Varies, often lower than epoxies
Water absorption	Low	Varies (polyesters: moderate, phenolics: low)	Varies (e.g., nylon: high, PP: low)
Ease of processing	Moderate, requires precise mixing and curing	Generally easier (polyesters: easy, phenolics: more complex)	High, often simpler processing methods
Recyclability	Not recyclable	Not recyclable	Recyclable
Cost	Moderate to high	Generally lower than epoxies	Varies, generally lower than thermosets

**Table 2 polymers-16-02152-t002:** Main properties for the different CNFs.

Property	CNFs [54]	CNFs [73]	VGCNFs [74]	VGCNFs [75,76,77]	VGCNFs [72]	VGCNFs [78]
Process	Electrospun	FC	-	-	FC	Gas-phase flow catalytic method
Diameter [nm]	106	60–150	50–200	150	20–80	200
Length [μm]	-	30–100	50–100	15 (10–20)	>30	10–20
Tensile modulus [GPa]	4806	400	240	516.5 (273–760)	-	-
Tensile strength [MPa]	206	2700	2920	3100 (2700–3500)	-	-
Strain to break [%]	1.46	1.5	-	-	-	-
Density [g/cm^3^]	-	1.8	2.0	2.0	>1.97	2.1
Thermal conductivity [W/m·k]	-	20	1950	-	-	-
Electrical resistivity [Ω·cm]	-	-	1 × 10^−4^	-	1 × 10^−2^	-

## Data Availability

Not applicable.

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
