# Peer review of "The Effect of Carbon Nanofibers on the Mechanical Performance of Epoxy-Based Composites: A Review"

_polymers, 2024, doi:10.3390/polym16152152_

Round 1
Reviewer 1 Report
Comments and Suggestions for Authors
This is a useful paper that summarizes studies of the mechanical properties of carbon nanofiber, epoxy-based nanocomposites, and laminate composites made of carbon nanofibers. However, there is a lack of in-depth discussion based on appropriate scientific explanations for mechanical properties regarding changes in the contents of carbon nanofiber. The paper would be better if authors supplemented the insufficient explanation with some organized research data. Words that need to be rechecked or explanations that need to be added have been summarized in the following comments.
1. Please check the word “Interlaminar shear (ILSS)” in the Abstract on page 1.
2. “Composite materials offer numerous advantages…” in the Introduction on page 1. Many advantage features are listed. It seems like a clearer and easier-to-read classification is needed. For example, pure physical properties, functionality, applicability, etc.
3. “Incorporating CNFs into nanocomposites offers several…” on page 2. Could you please add an explanation, including appropriate scientific evidence, as to why the weight reduction and maximization of mechanical properties of the part results in increases in modulus, strength, fracture toughness, fatigue strength, delamination resistance, impact strength, and structural damping?
4. VGCNF is first mentioned on page 4. Please mention it’s full name
5. Table 1 lists only the mechanical properties of CNF fabricated by electrospun and FC methods. If possible, it would be better to compare the mechanical properties of CNFs fabricated by gas phase flow catalytic method, PECVD method, and Arc-discharge deposition method, also.
6. On page 6, “these improvements depend on several factors, such as the aspect ratio of CNFs, the dispersion and distribution quality, alignment, the adhesion and interface…” It is explained that many features of the part are enhanced. Could you please add a scientific explanation? Or, it might be a good idea to add it to the discussion section.
7. CFNs are mentioned on page 6 and page 7. Unless there is a typo, please write the full name.
8. On page 7, please add an explanation of changes in mechanical properties, including scientific logic, regarding the correlation between the research results on bending strength or bending modulus and the CNFs content. Or please add it to the discussion section.
9. Use the same figure description for Figure 2 and Figure 3 on page 9. It would be good to explain in detail the graphs for nanocomposite or laminates separately. In Figure 4 and Figure 5, please additionally explain the figure description.
10. In section 3.3, “Effect of CNFs on stress relaxation and creep behavior”, please add an explanation about the relationship between CNFs content and stress relaxation.
11. In 3.4 section “Effect of CNFs on the interlaminar shear strength (ILSS)” Please add an explanation about the relationship between CNFs content and ILSS.
Author Response
Answers to reviewer 1's questions and comments, attached.

Reviewer 2 Report
Comments and Suggestions for Authors
1. Please provide some quantitative results on the mechanical properties of composites in the abstract section, and further suggestions on the mechanism of the influence of carbon nanofillers on the mechanical properties of composites should be revealed.
2. Introduction, the following questions should be addressed. 1) The performance advantages of epoxy resins compared with the other thermosetting resins and thermoplastic resins should be further highlighted and emphasized. 2) The dispersion of nanofillers in the epoxy resin and the mechanism of affecting the performance of the epoxy resin should be further summarized. 3) The latest research status of the influence of nano-fillers on the properties of epoxy resins should be analyzed and summarized briefly. It is recommended to review the relevant work below and make necessary supplements, such as Nanomaterials, 2021, 11, 1234. Polymer-Plastics Technology and Materials, 2022, 61(7): 709-725. Construction and Building Materials, 2024, 429: 136455.
3. When carbon nanofibers are applied to epoxy resins, perhaps the cost is a major issue that needs to be considered in a large scale applications. It is suggested that the corresponding modifications of the comments in this section be taken into account in Part 2.
4. For the preparation process of carbon nanofibers, it is suggested to add the typical preparation process schematics.
5. In part 3.1, the authors summarize the effects of CNF on the mechanical properties of epoxy resin composites. It is further suggested that the authors add schematic diagrams about the mechanism of the effect of CNF addition on the properties of epoxy resins.
6. For the strain rate response, why do you analyze the strain rate response as a separate part? In fact, it should be part of the mechanical properties. Relevant explanations are necessary.
7. Similar analysis and discussion of related mechanisms should be added to Section 3.3 and 3.4.
8. It is suggested that the conclusion part should be highly condensed and summarized.
Author Response
Answers to reviewer 2's questions and comments, attached.

Round 2
Reviewer 1 Report
Comments and Suggestions for Authors
Dear Authors,
the revised manuscript was improved well according to the reviewer’s comment.
It is recommended to be published as is.
Author Response
Thank you very much for your review and timely comments.
Reviewer 2 Report
Comments and Suggestions for Authors
Although the authors provided a revised manuscript, the following questions should be further answered.
1. For the question (1): The mechanism of the influence of carbon nanofillers on the mechanical properties of composites should be revealed. The revised manuscript does not contain this part of the information.
2. For the question (2): 1) The performance advantages of epoxy resins compared with the other thermosetting resins and thermoplastic resins should be further highlighted and emphasized. 2) The dispersion of nanofillers in the epoxy resin and the mechanism of affecting the performance of the epoxy resin should be further summarized.
The suggested research work should be further reviewed to prove the performance advantages of epoxy resins compared with the other thermosetting resins and thermoplastic resins and analyze the influence of nano-fillers on the properties of epoxy resins.
3. For the question (3): Whether this paper focuses on the cost of nanofillers or not, however, the problem of materials is always a very important issue of engineering application and popularization. As the review paper,it is suggested to add some related analysis and summary .
4. For the question (4): If the authors can’t give some schematic diagrams, it is also suggested to briefly describe the preparation process of carbon nanofillers.
Author Response
Answers to the reviewer's comments attached

Round 3
Reviewer 2 Report
Comments and Suggestions for Authors
accept